# Loss of accumulation zone exposes dark ice and drives increased ablation at Weißseespitze, Austria

Lea Hartl[1, 2], Federico Covi[3], Martin Stocker-Waldhuber[1], Anna Baldo[1, 4], Davide Fugazza[4], Biagio Di Mauro[5], and Kathrin Naegeli[6]

[1]Institute for Interdisciplinary Mountain Research, Austrian Academy of Sciences, Innrain 25, 6020 Innsbruck, Austria
[2]Alaska Climate Research Center, University of Alaska Fairbanks, 2156 Koyukuk Drive, Fairbanks, AK 99775, USA
[3]British Antarctic Survey, Cambridge, United Kingdom
[4]Department of Environmental Science and Policy, Università degli Studi di Milano, Milan, Italy
[5]Consiglio Nazionale delle Ricerche - Istituto di Scienze Polari, Milan, Italy
[6]Department of Geography, University of Zurich, Zurich, Switzerland

**Correspondence:** L. Hartl (lea.hartl@oeaw.ac.at)

**Abstract.** In recent years, firn and summer snow cover has decreased on Alpine glaciers, exposing larger areas of ice at higher elevations. This reduces albedo and leads to increased melt. To understand mass loss in former accumulation areas under conditions of rapid glacier recession, it is important to constrain the possible range of ice albedo in newly firn free regions, the duration of ice exposure, and the albedo-ablation relationship. We combine data from an on-ice weather station (3492 m.a.s.l.), ablation stakes, and remote sensing derived albedo to provide an overview of albedo and ablation in the summit region of Weißseespitze, the high-point of Gepatschferner (Austria), from 2018 to 2024. Before 2022, low albedo (<0.4) occurred on 3 to 8 days per year. In 2022, 37 days of low albedo values were recorded by the weather station and albedo dropped below previously observed minima of around 0.30 to values similar to those of the surrounding rock. Albedo remained very low in 2023 and 2024. Ice ablation at the stakes generally increased with the duration of ice exposure, reaching up to -1.7 m w.e. in high-melt years. Sensitivity experiments indicate that a five day period of very low albedo conditions (<0.20) results in about 30% more modeled surface melt if it occurs in late July compared to early September, highlighting temporal variability in the impact of ice exposure. The unique Weißseespitze dataset provides a starting point for further studies linking causes and effects of albedo changes in former accumulation zones.

## 1 Introduction

Glaciers outside the polar regions are losing mass at unprecedented rates (e.g., Hugonnet et al., 2021; Jakob and Gourmelen, 2023; The GlaMBIE Team, 2025), with projected mass losses of up to 41% by 2100 globally (Rounce et al., 2023) and up to 94% in the Alps (Zekollari et al., 2019), where complete glacier loss is anticipated regionally in the coming decades (e.g., Hartl et al., 2025).

During recent extreme melt seasons, repeated heat waves contributed to extreme glacier mass loss in the Alps (e.g., Zappa and Kan, 2007; Thibert et al., 2018; Cremona et al., 2023; Voordendag et al., 2023) and glacier thinning was observed in the highest reaches of Alpine glaciers (Berthier et al., 2023; Hartl et al., 2024). As snow lines rise and multi-year firn is depleted, bare ice becomes exposed. The newly exposed ice surfaces have a lower albedo than snow and firn, which affects the surface energy balance by increasing the amount of absorbed solar radiation, thereby creating a positive melt-albedo feedback. Albedo correlates strongly with glacier mass balance in the Alps and other mountain regions, highlighting the importance of this feedback for glacier evolution (e.g., Dumont et al., 2012; Zhang et al., 2021; Di Mauro and Fugazza, 2022).

As global glacier recession accelerates (e.g., Hugonnet et al., 2021; The GlaMBIE Team, 2025), overall glacier albedo and the albedo of bare ice have increasingly come into focus. At local and regional scales, ice albedo depends on meteorological factors like solar elevation, cloudiness, radiation budget and on surface roughness (e.g., Irvine-Fynn et al., 2014; Volery et al., 2025). Additionally, the presence of liquid water on the ice surface, the characteristics of the pore space and the weathering crust impact ice albedo (e.g., Dadic et al., 2013; Traversa and Di Mauro, 2024). Light absorbing impurities of organic and inorganic origin, including black carbon, algae, and dust, can lead to albedo reductions and darker glacier surfaces (e.g., Oerlemans et al., 2009; Gardner and Sharp, 2010; Qu et al., 2014; Gabbi et al., 2015; Zhang et al., 2017; Di Mauro et al., 2017; Goelles and Bøggild, 2017; Hotaling et al., 2021; Xiao et al., 2023).

Spaceborne remote sensing observations have been used to assess regional trends and seasonality of glacier albedo (e.g., Naegeli et al., 2019; Fugazza et al., 2019; Gunnarsson et al., 2021; Marshall, 2021; Shaw et al., 2021; Williamson and Menounos, 2021; Di Mauro and Fugazza, 2022; Traversa and Di Mauro, 2024). Some of the driving processes of albedo variability can be detected in remote sensing data, for example changes in debris cover, deposition of light absorbing particles related to dust or volcanic eruptions, and the presence of algae (e.g., Casey and Kääb, 2012; Azzoni et al., 2016; Di Mauro et al., 2017; Yue et al., 2020; Di Mauro et al., 2020; Gunnarsson et al., 2023). Several studies have shown a negative trend in ice albedo, or "glacier darkening", in the Alps (Naegeli et al., 2019; Fugazza et al., 2019; Di Mauro and Fugazza, 2022); however, the magnitudes of trends differ between regions.

Satellite derived albedo products have been used to force glacier energy balance models (e.g. Gunnarsson et al., 2023), but typically do not fully resolve small scale albedo variability (e.g. Hartl et al., 2020; Rossini et al., 2023). Remote sensing time series are often discontinuous due to limitations related to cloud cover and the frequency of satellite overpasses. Large scale reanalysis products and modeled albedo can show large discrepancies with observed albedo over glacier surfaces (e.g. Draeger et al., 2024).

Accordingly, in situ data are key as ground truth for remote sensing products and can complement temporally sparse remote sensing time series as well as contribute to calibration and validation exercises (e.g. Di Mauro et al., 2024). Continuous in situ albedo measurements on glacier surfaces remain rare (e.g. Ren et al., 2021) and long-term monitoring sites are essential for assessing the potential range of bare ice albedo and how this may change over time. Such information is also needed to improve albedo parametrization schemes in energy balance modeling, which often struggle to reproduce the observed spatial and temporal variability of glacier albedo (e.g. Brock et al., 2000, 2006; Collier et al., 2013; Eidhammer et al., 2021).

The continued loss of firn and reduced seasonal snow cover increase the relative importance of bare ice albedo and its
variability for glacier-wide albedo. In addition to impacts on glacier-wide mass balance, the exposure of bare ice at high
elevations has important implications for mass balance gradients, which play a key role in large-scale glacier modeling and
future projections (e.g. Miles et al., 2021; Schuster et al., 2023). Paul et al. (2005) described the impact of reduced summer
snow cover on albedo and mass loss during the exceptionally warm summer of 2003 in Switzerland. They noted that the loss
of snow cover at high elevations can lead to an inverted mass balance profile, pointing out the need for assimilation of albedo
observations in mass balance modeling.

In this study, we explore a unique, multi-scale dataset of in situ meteorological observations, remote sensing derived albedo,
and a network of ablation stakes from the highest region of Gepatschferner (Gepatsch Glacier), Austria's second-largest glacier.
We provide a quantitative overview of bare ice albedo in the summit region and (former) accumulation zone for recent years
(2018-2024). We focus on snow free periods and interannual albedo variability in context with ablation measurements and the
sensitivity of surface mass balance to albedo. We aim to determine the range of ice albedo that occurs in the highest reaches of
the glacier and to what extent the observations from the stake network can be linked to albedo and the duration of ice exposure.
We compare in situ data and Sentinel-2 derived albedo, assessing whether the satellite imagery captures short-term albedo
variations and the seasonal variability of melt patterns. Finally, we consider the sensitivity of modeled surface mass balance to
observed albedo. The overarching objective is to contribute to a more comprehensive understanding of ongoing high elevation
glacier mass loss and the processes driving it, with an emphasis on the extreme summer of 2022 and the years since.

## 2 Data and Methods

### 2.1 Study site and instrumentation

Weißseespitze (WSS, 3518 m.a.s.l.) is a glacierized peak in the Ötztal range in western Austria and forms the high point of
Gepatschferner (Fig. 1). Gepatschferner is Austria's second largest glacier and covered an area of 15.5 km$^2$ in 2017 (most recent
inventory, Helfricht et al. (2024)). Gepatschferner is retreating rapidly, with high loss rates particularly in the lowest sections
of the glacier tongue (Hartl, 2010; Hartl and Fischer, 2014; Stocker-Waldhuber and Kuhn, 2019; Piermattei et al., 2023). In
recent years, substantial losses have also been observed at higher elevations, in the (former) accumulation zone (Hartl et al.,
2025). This is in line with trends towards increasingly negative annual mass balance at three World Glacier Monitoring Service
(WGMS) reference glaciers in close proximity to Gepatschferner (Hintereisferner, Vernagtferner, Kesselwandferner). Annual
mass balance at these sites is strongly correlated with summer mass balance and the trend in ablation dominates the overall
mass balance trend, while winter mass balance at the reference glaciers with winter measurements shows no clear trend over
time WGMS (2025).

The summit region of WSS and Gepatschferner, above approximately 3480 m a.s.l., forms a small, dome-shaped cap. Ice
core analysis indicates that current surface ice at the site formed prior to the 1960s, likely in the pre-industrial era, and that the
age of the ice continuously increases with depth, reaching an age of $5.9 \pm 0.7$ ka (calibrated years before present; cal BP) just
above bedrock (Bohleber et al., 2020; Spagnesi et al., 2023). The ice in the summit region is cold based and ice thicknesses

in the range of 8 m to 14 m were measured in the central part of the summit "ice cap" in 2018 (Fischer et al., 2022; Stocker-Waldhuber et al., 2022b). Accordingly, an age range spanning millennia is condensed into only a few meters of ice at this location.

In the following, we focus mainly on the summit region around WSS (Fig. 1) and the (former) accumulation area of Gepatschferner, which consists of a wide, low-angle basin above about 3100 m.a.s.l. Additionally, we computed glacier-wide albedo statistics to further contextualize the data from the summit region and accumulation area.

On October 31, 2017, an automatic weather station (AWS) was installed near the summit of WSS at 3492 m a.s.l. (Fig. 1). The location of the AWS is in the highest region of the glacier within the (former) accumulation zone. Multi-year firn is no longer present around the AWS and bare ice is exposed if the seasonal snow cover does not persist through the summer. The station mast is installed in the ice and has been redrilled periodically depending on ice ablation to ensure the stability of the mast and roughly consistent sensor height above the ice surface. The AWS records standard meteorological parameters - air temperature and humidity (Rotronic-HC2S3), air pressure (Vaisala PTB110), wind speed and direction (Young-05103-45), up- and downwelling short- and longwave radiation (Hukseflux-NR01) - and distance to the surface (Campbell Scientific SR50a). The sensor acquisition frequency is one observation per minute and data are logged as 10 minutes averages. AWS data are available on the pangaea repository along with further information available parameters and sensor specifications (Stocker-Waldhuber et al., 2022a). Data are added to the parent repository in annual intervals. The SR50 records contain considerable noise and a data cleaning procedure was applied to extract a time series of smoothed surface height change (refer to Fig. S1 and S2 in Section 1 of the supplementary material for more information on this).

An automatic camera was installed on January 31, 2018, on a rock outcrop about 130 m east of the AWS. The field of view of the camera encompasses the glacier surface between the camera and the AWS (Fig. 1, panel e). The camera takes a picture every two hours between 8:00 and 18:00 local time. The period of record of the AWS and the camera extends from the respective date of installation through present. Figure S3 in the supplement gives an overview of the availability of different data types used in this study.

## 2.2 Ablation stakes

Seven ablation stakes were drilled on the Weißseespitze summit ice cap between 2017 and 2019. Another stake was added in 2022. The stakes consist of 2 to 2.4 m long wooden poles connected to each other with tubing to achieve total stake lengths of 6 to 12 m. At the time of installation, the stakes were drilled to be roughly plane with the ice surface; they emerge as ice ablation progresses. The maximum distance between the stakes is about 130 m and they are positioned at elevations between 3484 and 3502 m a.s.l. (Fig. 1, Fig. 2) The period of record for the stakes varies from one to seven summer seasons (Fig. S3). Stake readings were carried out in irregular intervals two to four times per year. Observations include snow depth at the stake locations (if any) and changes of the ice surface relative to the stake. In most cases, the latter is a negative value referring to a decrease in surface elevation due to ice ablation. At some stakes and for some observational periods, the formation of superimposed ice caused slightly positive surface change. Following Geibel et al. (2022), we assumed a reading uncertainty of $\pm 5$ cm for ice ablation measured at the stakes. We used a density of $900\,\mathrm{kg\,m^{-3}}$ to convert observed changes in the elevation of

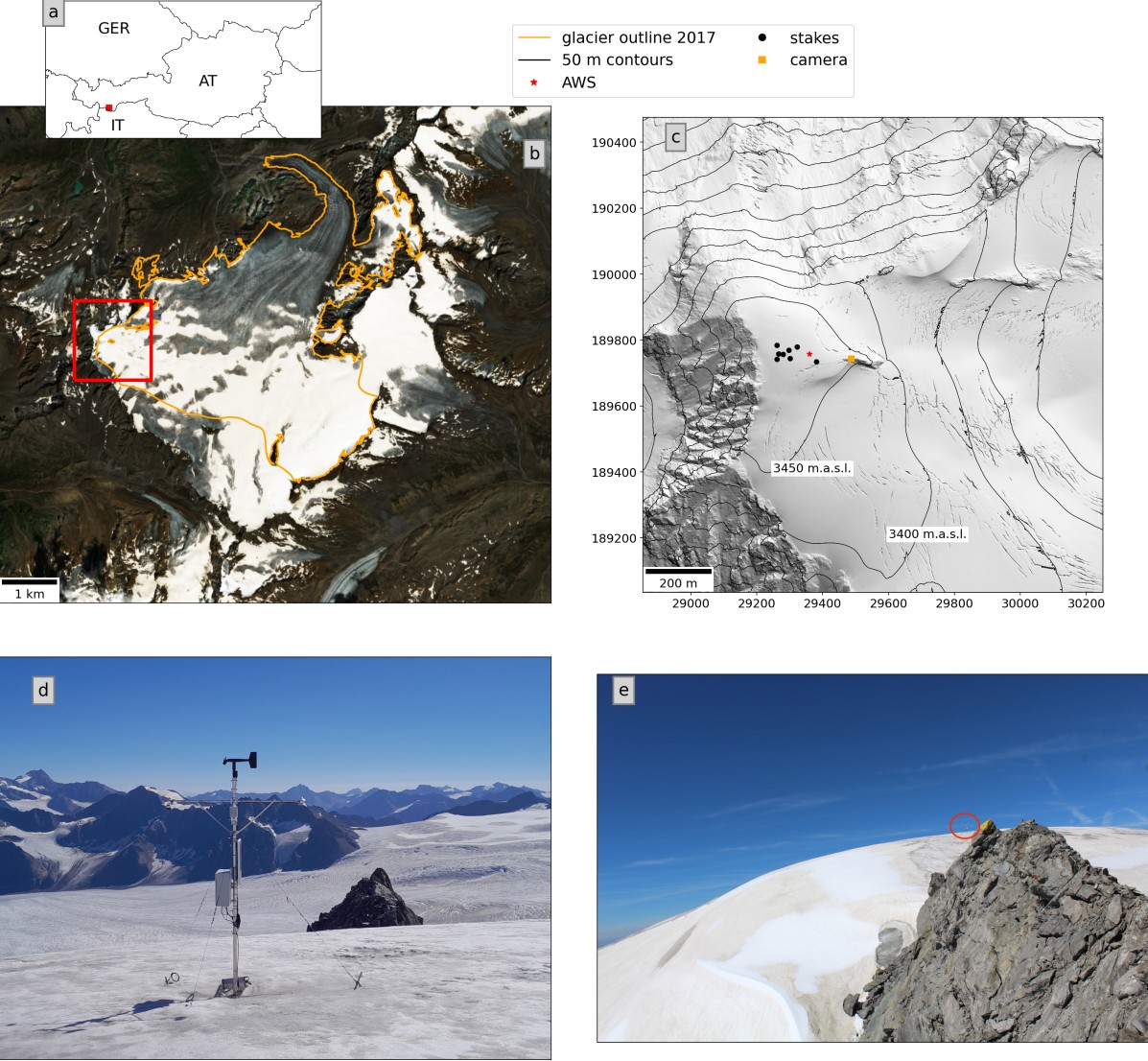

**Figure 1.** Panel a: Location of the study site in Austria. Panel b: Gepatschferner as seen in a Sentinel-2 scene acquired on 2021-08-14 with glacier outlines for Gepatschferner from the most recent regional glacier inventory (2017, Helfricht et al. (2024)). The red box indicates the subset shown in panel c. Panel c: Hillshade of the summit region with locations of the AWS, camera, and stakes. EPSG 31254, grid in metres. The hillshade and 50 m contour lines were derived from a digital elevation model for 2017 available from the geodata portal of the Tyrolean government (www.tirol.gv.at); elevations refer to orthometric heights. Panel d: The AWS as seen during a site visit on August 19, 2023 (Photo: M. Stocker-Waldhuber). The automatic camera is mounted on the rock outcrop visible behind the AWS. Panel e: Image taken by the automatic camera on the same day. The red circle indicates the position of the AWS.

the ice surface to mass change (e.g. Cogley, 2011). Snow density was not measured and unless otherwise stated the following analyses pertain only to ice ablation. Winter mass balance is not systematically measured at the site.

Stake positions were typically recorded with a Topcon HIPER V GNSS (Global Navigation Satellite System) receiver during the stake readings. Standard post-processing was applied to correct the raw data with the known location of reference stations. Horizontal movement of the stakes due to ice flow is not apparent, in line with expectations for the site and previous findings (Bohleber et al., 2020). We defined the positions of the stakes as the centroids of the GNSS coordinates recorded at the respective stakes between July and October over the years 2017 to 2024 (Fig. 2).

## 2.3 In situ albedo observations

Up- and downwelling shortwave radiation was measured with a Hukseflux-NR01 four component radiometer, which includes two Hukseflux-SR01 sensors for the shortwave components. The SR01 is a second-class thermopile pyranometer as per the ISO 9060-1990 standard, with a field of view of 180°. Shortwave radiation from the up- and down-facing sensor pair (incoming: $SW_{in}$, reflected/outgoing: $SW_{out}$) was used to compute daily albedo values, i.e. a daily representation of bi-hemispherical reflectance (BHR, Schaepman-Strub et al., 2006). To minimise the influence of variations in illumination angle, daily albedo was calculated as the sum of reflected shortwave radiation per day divided by the sum of incoming shortwave radiation per day. To eliminate zero offset effects during night time, $SW_{in}$ and $SW_{out}$ values below 2 W m$^{-2}$ were excluded from the albedo computation. In the following, "AWS albedo" refers to the daily BHR value as described above.

External factors affecting sensor performance and albedo are mainly related to snow or rime ice accumulation on the sensors. Riming is most common in winter during and immediately after storm periods. Instances of erroneously low albedo values in the winter months, when winter snow cover was present at the AWS, were filtered based on thresholds and visual inspection of the camera imagery. Unrealistically low albedo values occur when, e.g., the downward facing sensor is fully or partially rime covered. Unrealistically high albedo values >1 can occur when snow or rime accumulates on the upward facing sensor, due to the slope and aspect of the underlying surface (e.g., Weiser et al., 2016; Picard et al., 2020; Bohn et al., 2025), a non-perfect cosine response of the upward-facing sensor, or surface roughness effects related to, for example, wind-produced features in the snow. Manual quality control was carried out for outlier data points by checking weather conditions in the camera images.

The instrument manual for the Hukseflux-SR01 (Hukseflux, 2023) outlines the evaluation of measurement uncertainty of the SR01 in accordance with the Guide to the expression of Uncertainty in Measurements (GUM, ISO 98-3) and indicates "achievable" measurement uncertainties of 6.2% in summer and 9.9% in winter for daily totals. To account for possible aging issues since calibration, we assumed a relatively conservative uncertainty of 10% for daily totals of up- and downwelling shortwave radiation, respectively. Based on standard error propagation (root sum of the squares) for the ratio of up- and downwelling radiation, the resulting estimated uncertainty of daily albedo is 14%.

Radiation measurements on glaciers can be affected by surface roughness (e.g., Cathles et al., 2011; Irvine-Fynn et al., 2014), and changing or unknown angles of slope and sensor tilt (e.g. Abermann et al., 2014; Weiser et al., 2016). The glacier surface at the WSS AWS is slightly sloped in a northeasterly direction, at an angle of around 6°-8° degrees as per a digital elevation model (DEM) derived from an airborne laser scanning survey carried out in 2017 (https://www.tirol.gv.at/sicherheit/geoinformation/geodaten-tiris/laserscandaten/, last access: 24 October 2024). Issues related to the sensitivity of daily AWS albedo to surface tilt angle are discussed in greater detail in the supplementary material (supplement Section 2, Fig. S4, S5). The estimated effect of the

slope angle on albedo is an order of magnitude lower than the uncertainty associated with sensor specifics (14%, see above) and we therefore neglected more sophisticated corrections. We note that such corrections should be considered for assessments of sub-daily albedo (e.g., Weiser et al., 2016).

In the following, we refer to albedo below 0.40 as "low" and albedo below 0.20 as "very low". These categories are roughly aligned with the 5% and 1% quantiles of the albedo time series. The threshold of 0.40 follows approaches by Fugazza et al. (2019) and Di Mauro and Fugazza (2022), who used this value to discriminate ice and snow. Glacier ice albedo below 0.20 can occur when the surface is loaded with organic and inorganic impurities or partially debris covered, and has been linked to the presence of liquid water (e.g., Paul et al., 2005; Di Mauro et al., 2017; Di Mauro, 2020; Rossini et al., 2023).

## 2.4    Sentinel-2 derived albedo

We used the Sentinel-2 (S2) harmonised surface reflectance product (Level-2A, Drusch et al., 2012) to generate broadband surface reflectance estimates for the summit region of Weißseespitze and the overall glacier area of Gepatschferner. S2 data at 10 m x 10 m resolution were obtained through Google Earth Engine (GEE, Gorelick et al., 2017) for 2018 to 2024. The SWIR 1 and SWIR 2 bands were resampled from 20 to 10 m resolution using nearest neighbor resampling during the GEE download.

The S2 Scene Classification Layer band was used to remove pixels classified as saturated or defective. Cloud filtering was carried out using the Cloud Score+ product with a threshold of 0.5 in the cloud score band (Pasquarella et al., 2023).

Multi-spectral reflectance was converted to broadband reflectance using the conversion developed by Liang (2001), where broadband reflectance is a function of the blue (B2), red (B4), NIR (B8), SWIR1 (B11) and SWIR2 (B12) bands. We omitted corrections for the effects of reflectance anisotropy based on the results of Naegeli et al. (2017), who showed that the impact of

such a correction is minimal for glacier albedo derived with the above approach. We note that the derived quantity is not BHR but rather a hemispherical directional reflectance factor (HDRF, Schaepman-Strub et al., 2006). We use the term "S2-derived albedo" to refer to this in the following.

For a first-order estimate of uncertainty in the S2-derived albedo, we referred to uncertainties for the spectral bands of S2 L2A data given in (Gorroño et al., 2024, Table 2, MCM approach), which range from 1 to 3% in the NIR and SWIR bands and

from 8 to 10% in the visible bands for a test scene. Standard error propagation (root sum of the squares) yields an uncertainty of $\pm 13\%$ for S2-derived albedo assuming independent errors and $\pm 16\%$ when taking spectral error correlation (Gorroño et al., 2024, Fig. 5) into account. We used the latter value as an uncertainty estimate of S2-derived albedo in our analysis. We note that this is a strongly simplified approach compared to the comprehensive radiometric uncertainty assessments developed in, e.g., Gorroño et al. (2017); Graf et al. (2023); Gorroño et al. (2024) and point to these works for further background.

### 2.4.1    Point-scale S2-derived albedo

S2-derived albedo was extracted for the positions of the ablation stakes and the AWS for the period of record (2018-2024). To account for uncertainties in the stake positions, a 5 m buffer (Fig. 2) around the points was applied. The value for each point corresponds to the pixel whose centroid is within the buffer. If there were multiple pixel centroids in the buffer, the average over the pixels was taken. Manual quality control was carried out to remove outliers and unreasonably low values in winter

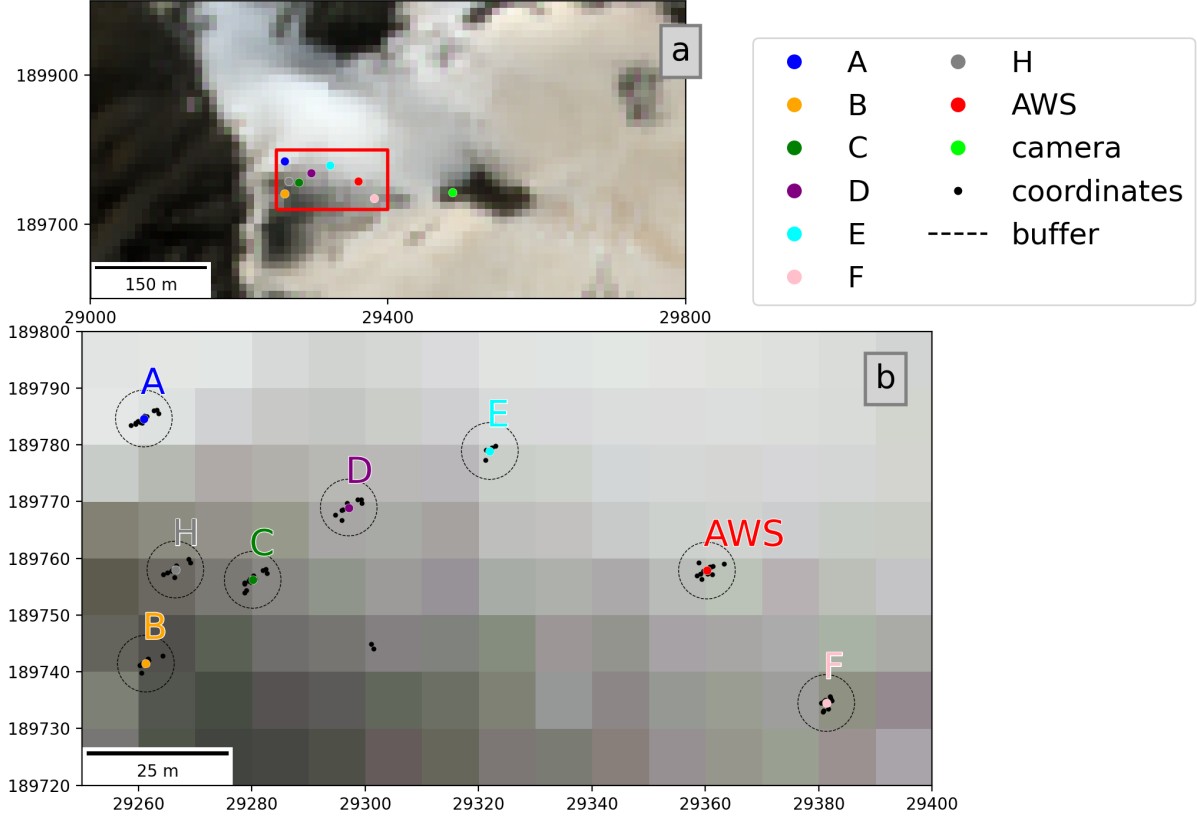

**Figure 2.** Positions of the AWS and the ablation stakes used in the following analyses plotted over a true color Sentinel-2 image acquired on 14 August 2021. Panel a: View of the summit area. The red box marks the subset shown in panel b. Panel b: Centroids of the stake positions (colored markers) with the GPS coordinates used to derive the centroids (black markers) and the 5 m buffer applied to extract the S2-derived albedo. All data are shown projected to the Austrian national grid (EPSG 31254, grid in meters).

by comparing the point data with true color composites of the wider area. 5.2% of the cloud-filtered S2 data were discarded during this process due to cloud shadows and clouds not caught in the filter.

To relate ablation at the stake positions with albedo during the respective measurement intervals, the number of "low albedo" days per interval at each stake was estimated by augmenting the relatively sparse S2 time series with AWS data. The underlying assumption was that once "low albedo" values below 0.40 are reached at a given stake, albedo remains low until fresh snow brightens the glacier surface, and that a given snowfall brightens the surface at all stake positions and at the AWS. Hence, low albedo periods at the stakes were considered to begin any time the S2-derived albedo dropped below 0.40 at a given stake and to end if albedo at the AWS increased by at least 0.20 from one day to the next, indicating a brightening of the surface due to snowfall. This resulted in an estimated number of low albedo days at each stake for each measurement interval between consecutive stake readings.

### 2.4.2 Conditions during seasonal minimum snow cover

For an assessment of S2-derived glacier-wide albedo, we manually selected cloud free S2 scenes of Gepatschferner during conditions of minimum snow cover for 2018 to 2024, choosing the image with the least amount of snow and clouds for each year. We computed broadband albedo as described above for all snow and ice pixels within the glacier boundary. Non-ice and snow pixels containing debris cover and areas along the edge of the glacier tongue where the ice has retreated since the glacier outline was mapped were masked based on band ratio filters of NIR/SWIR1 and Red/SWIR1, similar to the approach of Paul et al. (2016). The GEE hillshadow function was applied based on the Copernicus 30 m DEM to mask terrain shadows on the lowest section of the glacier tongue. Mean S2-derived albedo was computed for the entire glacier area and for 20 m elevation bands.

### 2.5 Sensitivity experiments

To assess the sensitivity of the energy available for melt to albedo variations, we used the Coupled Snowpack and Ice Surface Energy and Mass Balance Model in Python (COSIPY, Sauter et al., 2020) to simulate surface mass balance at the AWS location during periods of ice ablation between July and September. COSIPY requires air pressure, air temperature, relative humidity, incoming solar radiation, cloud cover or incoming longwave radiation, wind speed, precipitation, and optionally snowfall as input parameters. COSIPY version 1.4 was adapted to use albedo as measured at the AWS as an additional input parameter rather than applying the default albedo parameterization scheme.

We performed idealized sensitivity experiments to isolate the impact of albedo on melt in otherwise unchanged conditions by systematically varying albedo. The experiments were set up to highlight the influence of surface albedo on overall ablation, considering how "darker" vs. "brighter" bare ice influences surface mass balance, and how this varies with the time of year. To this end, we considered simplified summertime scenarios with a bare ice surface, omitting considerations of energy balance processes in a seasonal or multi-annual snowpack and energy input from precipitation. Precipitation was assumed to be zero in the experiments described below. The albedo of the ice surface was kept constant over time and varied in increments of 0.05 for each run, so that each model run corresponded to one albedo value between 0.05 and 0.95. For high albedo values that would be indicative of snow, the interpretation is that only a minimal amount of snow covers the bare ice, as would be the case for example during a small summer snow fall at high elevation. An ice thickness of 6 m and a bottom temperature of -4°C were set as initial conditions for all model runs.

Two experiments were performed: For a case-study focused on the 2022 season, we forced the model with meteorological input from a week-long heatwave in July 2022. For a more generalized assessment, the model was forced with "average" July-September conditions based on the 2018-2024 AWS time series. AWS records were grouped by time and day and averaged over all years. In both cases, we ran the model with the idealized, constant albedo as described above, and generated a control run with albedo as measured at the AWS. The model was run for the respective time periods (2022 heat wave, and 1 July to 30 September for average 2018-2024 conditions) without a spin-up phase. Model output consists of partitioned energy fluxes at hourly intervals.

To evaluate how well the model is able to reproduce observations, we compared modeled surface mass balance with surface height change extracted from the SR50 data during the July 2022 heatwave and an additional sub-period during the transition from snow to bare ice conditions in August 2021. The 2022 period was chosen to assess the time period of the experiment described above. The SR50 data is very noisy during this time, resulting in heavily filtered surface height data that do not resolve short term variations (refer to section 1 in the supplement for more information on the filtering process). The 2021 period was chosen to allow a more detailed assessment with better quality observations of surface height change during the transition from snow to ice melt and subsequent small snowfall events. For the evaluation run, snowfall derived from the SR50 data was provided to the model as an additional input parameter. Fig. S6 (section 3 in the supplement) shows a comparison of observed and modeled surface change. The initial phase of the evaluation period shows continuous snow and, subsequently, ice melt. Differences between modeled and observed surface height during this phase ranged from about 1 to 4 cm. Daily change rates agreed within $\pm 2$ cm and the timing of the transition from snow to ice melt was captured well ($\pm 1$ day). Discrepancies between observed and modeled surface height change increased around snowfall events, with differences in daily change rates of up to $\pm 5$ cm. Over the 26 day evaluation period, the root mean square error (rmse) of observed and daily change rates was 4.8 cm, with an R2 score of 0.6 and an absolute mean bias of 0.8 cm. Despite daily variability in change rates and model fit, modeled and observed cumulative surface change after 18 days was within $\pm 1$ cm. Snowfall then lead to positive surface height change, which was not captured well by the model (Fig. S6).

Overall, the evaluation run indicated satisfactory agreement of model and observations during melt conditions. It also highlighted the difficulties of capturing small summer snowfalls both observationally and in terms of representation in the model. By assuming a bare ice surface for our sensitivity experiments, we aim to avoid the uncertainties introduced by precipitation input and the related parameterizations. At the process level, this represents a major simplification. Nonetheless, based on the ability of the model to reproduce the observed height change during the transition from snow melt to a bare ice surface and during ice ablation, we suggest that meaningful conceptualizations of the impact of albedo on ice melt can be carried out with the simplified approach outlined above.

## 3 Results

### 3.1 Time series of broadband albedo at WSS

#### 3.1.1 In situ dataset

The glacier surface at the AWS location is snow covered most of the year and mean monthly albedo for 2018 to 2024 was around 0.80 from October to May. Mean monthly albedo dropped to 0.71 in June, 0.64 in July, and 0.53 in August and increased again to 0.69 in September (Tab. 1). The standard deviation of albedo was higher in July, August, and September (0.14-0.20) compared to the rest of the year ( 0.10). August had the lowest mean albedo and highest standard deviation. 5% of all daily albedo values in the time series are lower than 0.46 and 1% are below 0.21.

**Table 1.** Monthly mean albedo and standard deviation at the AWS, 2018-2024.

| Month | Mean | Standard deviation |
|-------|------|--------------------|
| Jan.  | 0.77 | 0.10 |
| Feb.  | 0.78 | 0.10 |
| Mar.  | 0.79 | 0.10 |
| Apr.  | 0.79 | 0.08 |
| May   | 0.80 | 0.10 |
| Jun.  | 0.72 | 0.09 |
| Jul.  | 0.64 | 0.14 |
| Aug.  | 0.53 | 0.20 |
| Sep.  | 0.69 | 0.17 |
| Oct.  | 0.76 | 0.11 |
| Nov.  | 0.78 | 0.10 |
| Dec.  | 0.81 | 0.10 |

Fig. 3 shows daily albedo compared to the time series mean. On average, a gradual albedo decline is apparent from late May until late August as seasonal snow ages or firn layers become exposed. A seasonal albedo minimum coinciding with a snow free ice surface at the AWS is typically reached during the second half of August. The subsequent increase in albedo is abrupt compared to the months-long decrease during snow melt, indicating the sudden, brightening influence of fresh autumn snow on the darker summer surface.

Annual minimum AWS albedo values were around 0.30 from 2018 to 2021 (Tab. 2, Fig. 3) and occurred during periods with little or no remaining snow cover around the AWS (Fig. S7, supplementary material). From 2022 onwards, the minima were considerably lower, dropping below the previously observed range to 0.17 in 2022 and 2023 and to 0.16 in 2024 (Tab. 2).

2022 additionally stood out as the year with the earliest minimum and the longest period of "low albedo" values to date with 37 days of albedo lower than 0.40, including five days of albedo below 0.20. In previous seasons, the number of days below 0.40 ranged from three (2018) to eight (2020) and days below 0.20 did not occur. In 2023 and 2024, five and six days of albedo below 0.20 were recorded, respectively (Tab. 2). The onset date of low and very low albedo conditions in 2024 cannot be determined due to an outage of the AWS in August; however, "very low" albedo conditions persisted into September, marking the latest recorded date of albedo values below 0.20 in the time series.

### 3.1.2 Sentinel-2 derived albedo

Comparing S2-derived albedo extracted for the AWS location with the in situ albedo shows good overall agreement (Fig. 4). The S2-derived albedo captures the pronounced "low albedo" phases related to exposure of bare ice and the brightening events from snow fall (e.g. August 2022, Fig. 5). The absolute mean bias and rmse of the S2-derived albedo compared to the in situ

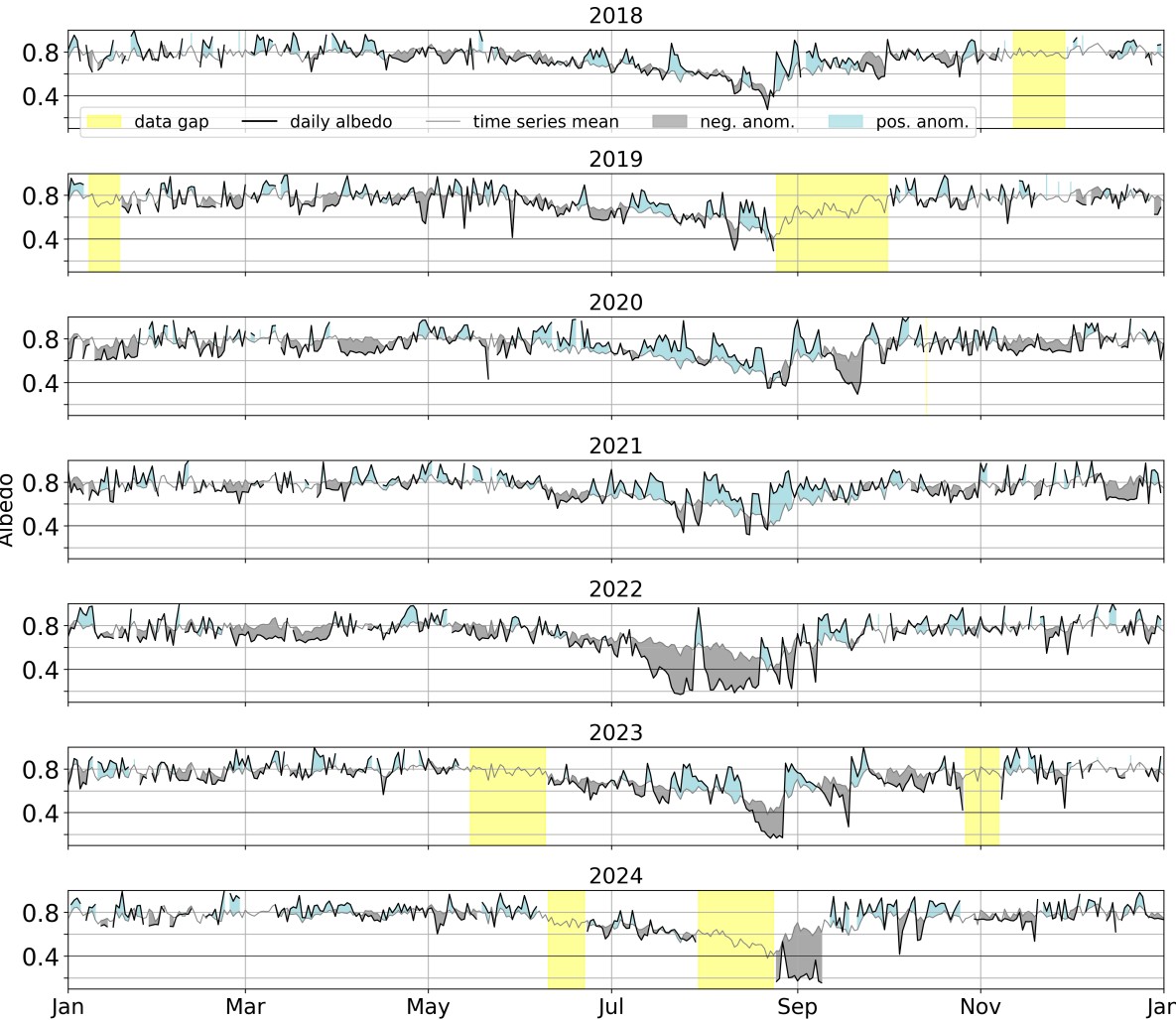

**Figure 3.** Daily albedo at the AWS for 2018-2024 (black line). Yellow shading marks major gaps in the time series due to instrumentation failures. The grey line indicates the time series average. The blue and grey areas indicate positive (blue) and negative (grey) deviations from the time series average over all seven years.

data were 0.074 and 0.095, respectively, for all 412 data points shown in Fig. 4. S2-derived albedo and AWS albedo agree well for the darkest bare-ice values around 0.20 with no apparent tendency for over- or underestimation (mean bias: 0.037, rmse: 0.041, N: 12), increasing our confidence that very dark bare-ice conditions in the WSS summit region were captured by the S2 acquisitions and not obscured by, e.g., mixed pixel effects.

S2-derived albedo at the stakes was greater than 0.56 in 90% of the cases shown in Fig. 5, and dropped below 0.30 in 5% of cases. "Low albedo" periods in the S2-derived time series generally coincided with "low albedo" periods at the AWS and were observed in July (2022), August (2018, 2019, 2022, 2023, 2024), and September (2024). In 2018, S2-derived "low albedo" was

**Table 2.** Number of days per year with albedo below 0.40 and 0.20 ("below 0.40" is inclusive of days below 0.20), as well as the annual minimum values and the dates when the minima occurred.

| Year | Days <0.40 ("low albedo") | Days <0.20 ("very low albedo") | Minimum | Date of minimum (mm-dd) |
|------|---------------------------|--------------------------------|---------|-------------------------|
| 2018 | 3 | 0 | 0.27 | 08-22 |
| 2019 | 4 | 0 | 0.29 | 08-24 |
| 2020 | 8 | 0 | 0.29 | 09-20 |
| 2021 | 5 | 0 | 0.32 | 08-16 |
| 2022 | 37 | 5 | 0.17 | 07-24 |
| 2023 | 11 | 5 | 0.17 | 08-23 |
| 2024 | 15 | 6 | 0.16 | 09-08 |

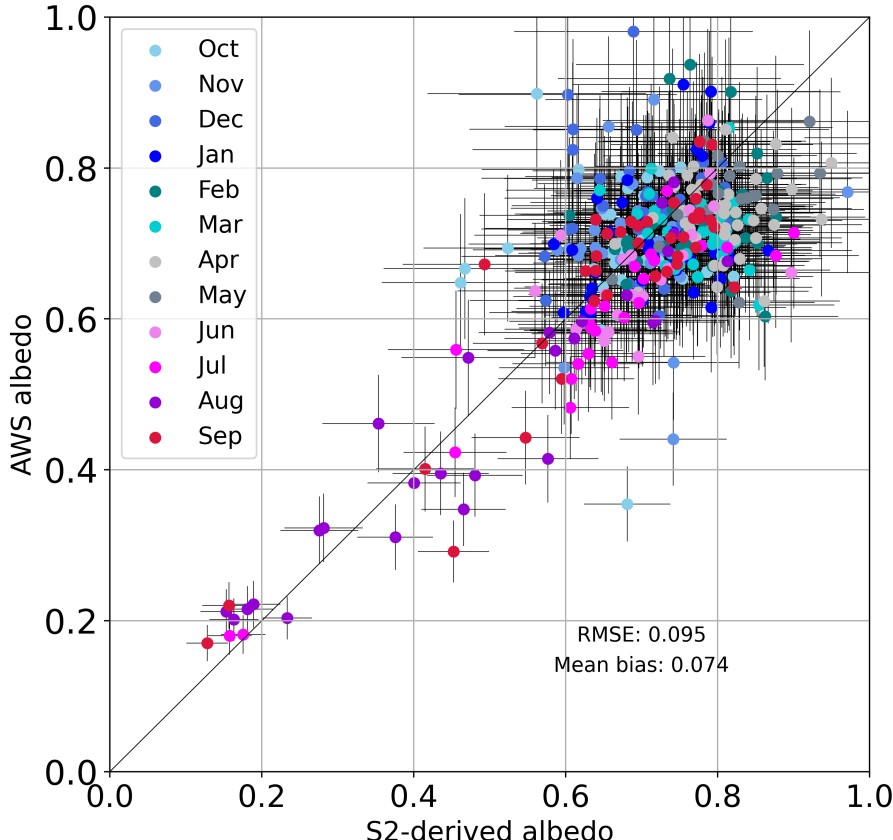

**Figure 4.** S2-derived albedo plotted against AWS in situ albedo from the same day, colour coded by month. Black lines denote error bars for the in situ and S2-derived albedo based on the estimated uncertainties of 14% and 16%, respectively.

observed five days prior to "low albedo" conditions at the AWS. The earliest date of S2-derived albedo below 0.40 was July 22, 2022, indicating a very early start of the bare-ice period in the WSS summit region in 2022 in line with AWS data.

"Low albedo" periods in the S2-derived albedo dataset often occur time synchronously at the various stake positions and the AWS location (Fig. 5). However, spatial variability of albedo in the WSS summit region is also apparent. The stakes are not expected to melt out at exactly the same dates due to varying snow melt patterns. In addition, the irregular nature of the S2 time series makes it challenging to meaningfully interpret shifts of a few days - these may be due to snow melting earlier or later at one stake compared to the next, or to differing availability of S2 imagery. For example, S2-derived albedo at stake B dropped to below 0.3 in the summer of 2021 while the other stakes and the AWS position retained considerably higher albedo values. In summer 2018, S2-derived albedo at the AWS position dropped to slightly below 0.40 only once. In contrast, most stakes had "low" and even "very low" albedo values for multiple S2 acquisitions, indicating a longer period of exposed bare ice at the stakes than at the AWS and considerable small-scale spatio-temporal variability of albedo.

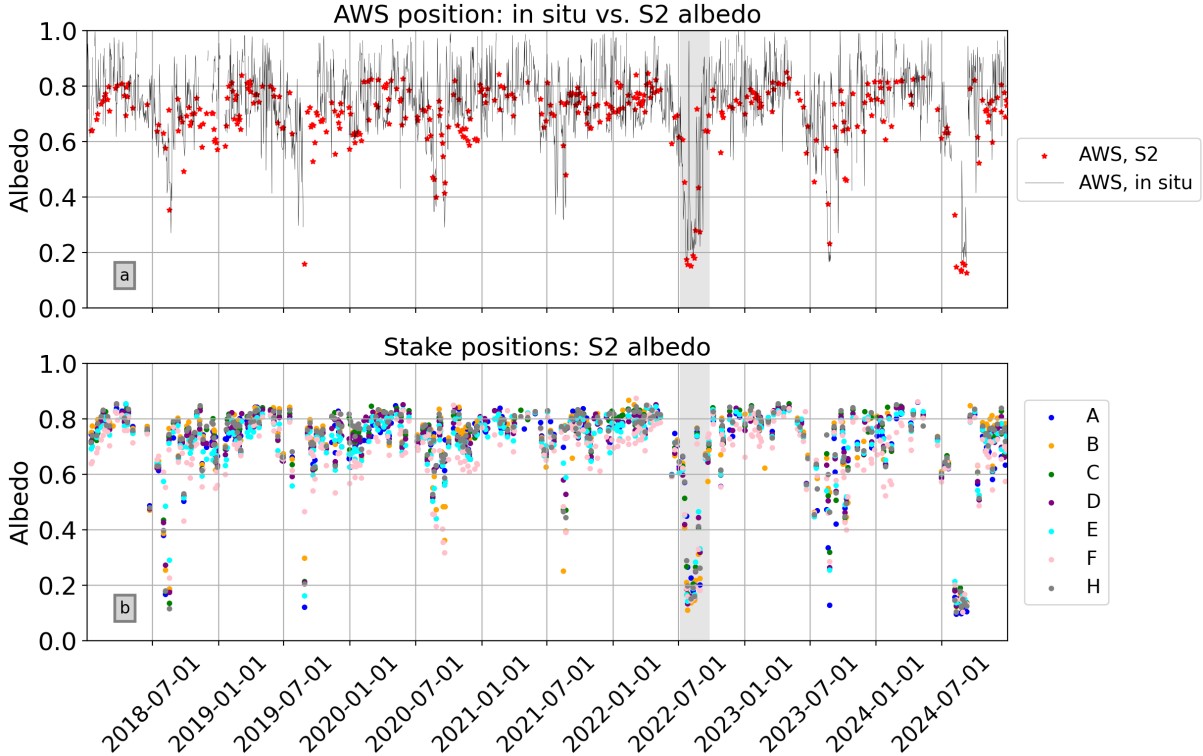

**Figure 5.** a) Daily albedo as measured in situ at the AWS (black line) and Sentinel-2 derived albedo extracted for the AWS position. b) Sentinel-2 derived albedo extracted for the stake locations. The grey shading indicates the temporal subset shown in more detail in Figure 7.

## 3.2 Albedo at the ablation stakes in the WSS summit region

Ablation varied considerably between different stake positions on the small summit ice cap (Fig. 1, 2). Cumulative ice ablation over the melt seasons 2018-2023 was 1701 mm w.e. at stake A, while stakes B, C, and F lost 2799, 3069, and 3555 mm w.e., respectively. Stake F has the longest period of record - August 6 2017 to September 20 2024. Cumulative ablation during this period was 5184 mm w.e. In the extremely warm summer of 2022, the four stakes mentioned above lost between 882 mm w.e. $yr^{-1}$ and 1521 mm w.e. $yr^{-1}$ of ice. During the more moderate summer of 2021, ablation values ranged from no ice ablation at stake A to 324 mm w.e. $yr^{-1}$ at stake B (Tab. 3).

Fig. 6, panel a, shows the estimated number of "low albedo" days per year at the stakes as derived from S2-albedo plotted against annual ablation. Estimated "low albedo" days per year range from zero to a maximum of 28 at stakes B and F in 2022 (Tab. 3). Annual ice loss tends to increase with the number of "low albedo" days. Conversely, low ablation values tend to occur in years with few "low albedo" days. The Pearson correlation coefficient for ablation against the estimated number of low albedo days as shown in Fig. 6, panel a, is -0.8.

2018, 2022, and 2024 stood out as years with numerous "low albedo" days and high ablation values at the stake positions. For example, the estimated number of "low albedo" days was 22 in both 2018 and 2022 at stake A. Annual ablation at this stake was also very similar in both years with -855 and -882 mm w.e. $yr^{-1}$, respectively. At stake B, the estimated number of "low albedo" days was 22 in 2018 but rose to 28 in 2022. Annual ablation at stake B was considerably higher in 2022 than in 2018 with -1521 mm w.e. $yr^{-1}$ compared to -738 mm w.e. $yr^{-1}$ (Tab. 3).

The correlation apparent in the data is in line with expectations. However, the data contain outliers (e.g., no ablation and 17 "low albedo" days at stake D in 2018) and show a considerable range of ablation values for a given count of "low albedo" days. For example, in cases with four estimated low albedo days, no ablation to losses greater than 650 mm w.e. were recorded depending on the stake and measurement period (Fig. 6, panel a, Tab. 3)).

The outlier value at stake D in 2018 can be investigated further: Stake readings taken on July 31 and August 29 indicate that there was no ice ablation at the stake position during this period. 97 cm of snow were recorded on July 31 at stake D, while neighboring stakes had zero remaining snow cover on this date. On September 27, 2 cm of positive ice surface height change were recorded at stake D. At stake C, the closest neighbor to stake D, the ice surface lowered by -85 cm (-765 mm w.e.) during the first period (July 31 to August 29). The second period (August 29 to September 27) saw a slightly positive surface elevation change (+4 cm) similar to stake D. Fig. S8 (supplement) shows impressions of the glacier surface in the WSS summit region during the stake readings on August 29, 2018. The key factor determining the diverging evolution at stakes D and C between July 31 and August 29 seems to have been the considerable snow pack that was present at stake D in late July. In an S2 scene acquired on August 5, a strong albedo gradient associated with the transition from snow to bare ice is apparent in the vicinity of stake D (Fig. S9; refer to Sec. 4.1.2 for further discussion).

Fig. 6, panel b, breaks the ablation stake data down into individual measurement periods, showing the mean ablation rate during each period plotted against the percentage of estimated "low albedo" days in the same period. Low mean ablation rates tend to occur in measurement periods with few low albedo days. The highest mean ablation rates were reached in the

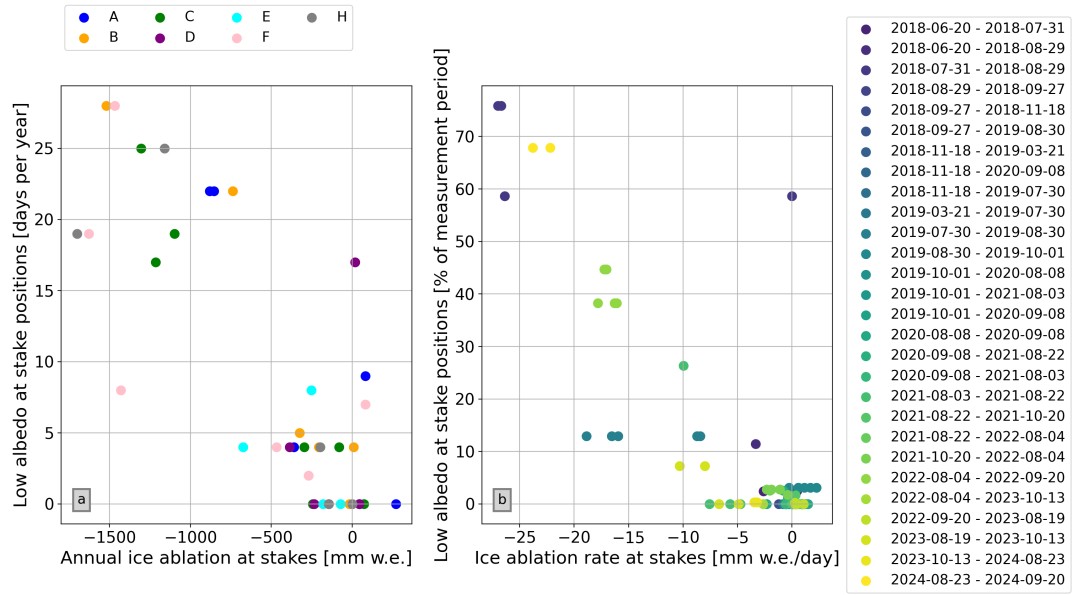

**Figure 6.** Panel a) Cumulative annual ablation at the stakes plotted against the estimated number of "low albedo" (<0.40) days per year at each stake position. Panel b) Ice ablation rate (ablation per measurement period divided by the length of the period) for each measurement period at the stakes plotted against the percentage of "low albedo" (<0.40) days at the respective stakes during the same measurement period.

measurement period with the highest amount of estimated "low albedo" days, in August 2018. Over 70% of the days between
the stake readings on July 31 and August 29, 2018, were "low albedo" days at the positions of stakes A and B and both stakes
reached mean ablation rates of over 25 mm w.e. per day. In 2024 (August 23-September 20), ablation rates between 22 and 24
mm w.e. per day were reached at stakes F and H, with 68% of this measurement period exhibiting low albedo days. August
4 to September 20, 2022, also stood out with high daily ablation rates between 16 and 19 mm w.e., although the percentage
of "low albedo" days during this period was lower (38%-44% depending on the stake). In August 2019, similar mean ablation
rates were reached at three stakes with only 12% of "low albedo" values during the measurement period.

**Table 3.** Estimated number of "low albedo" (<0.40) days derived from S2-data and the AWS time series and annual surface change (mm w.e.) at stakes A-H.

| Year | | A | B | C | D | E | F | H |
|------|------|------|------|------|------|------|------|------|
| 2018 | Low albedo days | 22 | 22 | 17 | 17 | 8 | 8 | 22 |
| | Ablation | -855 | -738 | -1215* | 18 | -252 | -1431* | / |
| 2019 | Low albedo days | 4 | 4 | 4 | 4 | 4 | 0 | 4 |
| | Ablation | -360 | 9 | -81 | -387 | -675 | 0 | -198 |
| 2020 | Low albedo days | 0 | 4 | 0 | 0 | 0 | 7 | 0 |
| | Ablation | 45 | -207 | 72 | 45 | -72 | 81 | 0 |
| 2021 | Low albedo days | 0 | 5 | 0 | 0 | 0 | 2 | 0 |
| | Ablation | 270 | -324 | -243 | -234 | -180 | -270 | -144 |
| 2022 | Low albedo days | 22 | 28 | 25 | 25 | 25 | 28 | 25 |
| | Ablation | -882 | -1521 | -1305 | / | / | -1467 | -1161 |
| 2023 | Low albedo days | 9 | 0 | 4 | 4 | 4 | 4 | 0 |
| | Ablation | 81 | -18 | -297 | / | / | -468 | -144 |
| 2024 | Low albedo days | 19 | 19 | 19 | 19 | 19 | 19 | 19 |
| | Ablation | / | / | -1098 | / | / | -1629 | -1701 |

* : Drilled in 2017. 2018 value includes ablation between last reading in 2017 and first reading in 2018.

### 3.3 A closer look at summer 2022

Summer 2022 stood out as the first unusually dark year in the albedo time series and the periods of "low" and "very low" albedo values were considerably longer than in all other years (Figs. 3, 5). We consider the "low albedo" period of 2022 in more detail to illustrate the characteristics of this exceptional season in the WSS summit region.

The AWS and S2 data show a relatively gradual decline of albedo from about 0.8 to under 0.20 from July 5 to July 22, 2022. This coincided with a period of increasing temperatures between July 7 and 22. Albedo at the position of stake A remained slightly higher compared to the other stakes due to lingering snow cover (Fig. 7 panel a, Fig. 8). Nighttime temperatures remained above freezing from July 13 to 28 (Fig. 7 panel b), when temperatures dropped and a small snowfall briefly brightened the surface. With the thin snow cover, albedo rose to over 0.9 at the AWS before dropping back to around 0.20 in early August.

During a stake reading on August 4, all stake positions were snow free. Albedo remained continuously low until another snowfall on August 19-20. This was followed by about five days of snow melt and darkening surfaces. Then, a series of small snowfalls followed by brief melt periods produced a rapid succession of low and relatively high daily albedo values. This short-term variation was not captured in S2 imagery due to cloud cover. In the second week of September, a more substantial snowfall brought a lasting brightening of the surface, marking the start of "snow season" in the summit region.

Ice surface height change at stakes A, B, C, F, and H between August 4 and September 20 ranged from -84±5 cm to -93±5 cm. Significant spatial variability of albedo between the stake locations was not apparent. The estimated number of days with "low albedo" values was between 18 and 21 depending on the stake. The satellite derived albedo at the stake locations suggests that stake A melted out a few days later than the other stakes (Fig. 7 panel a, Fig. 8) but this period was not captured by the stake readings. Considering the S2-derived albedo in Fig. 8, the ice of the WSS summit and the neighbouring ice-free areas

had similar albedo values between August 4 and 16, i.e. the ice was as dark as the surrounding rock.

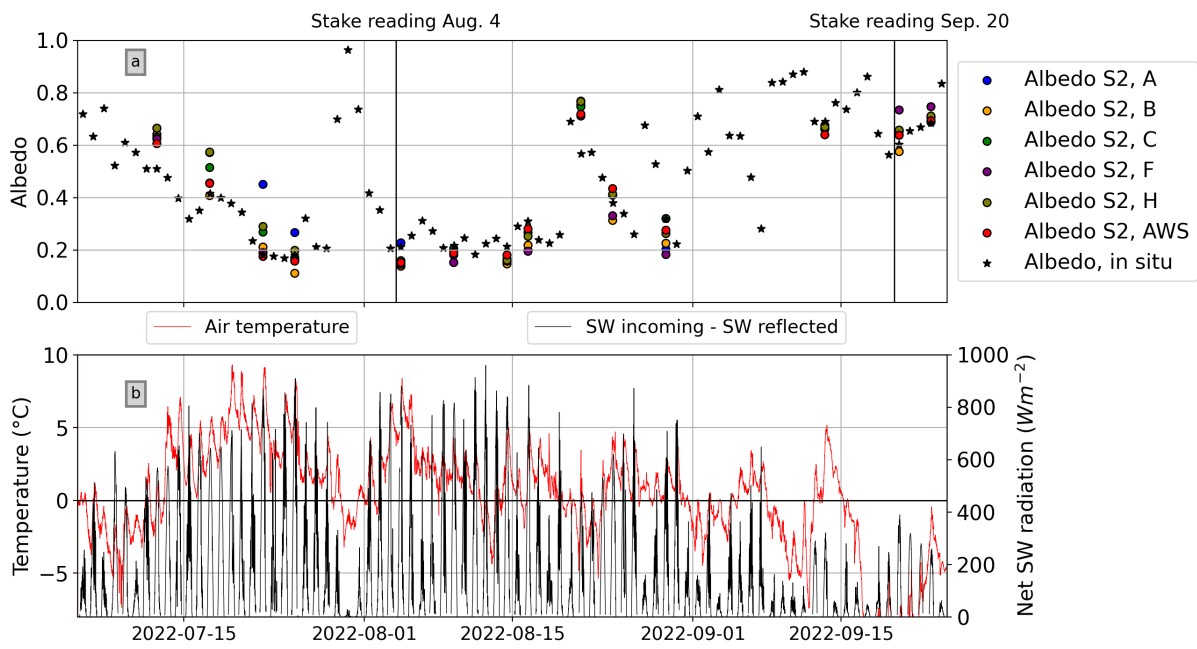

**Figure 7.** Panel a) Daily albedo at the AWS (black stars) from 2022-08-01 to 2022-09-01, and S2-derived albedo from cloud-free acquisitions during the same time period for the stakes shown in panel a. Panel b) Air temperature (red line) and net shortwave radiation (incoming - reflected) at the AWS.

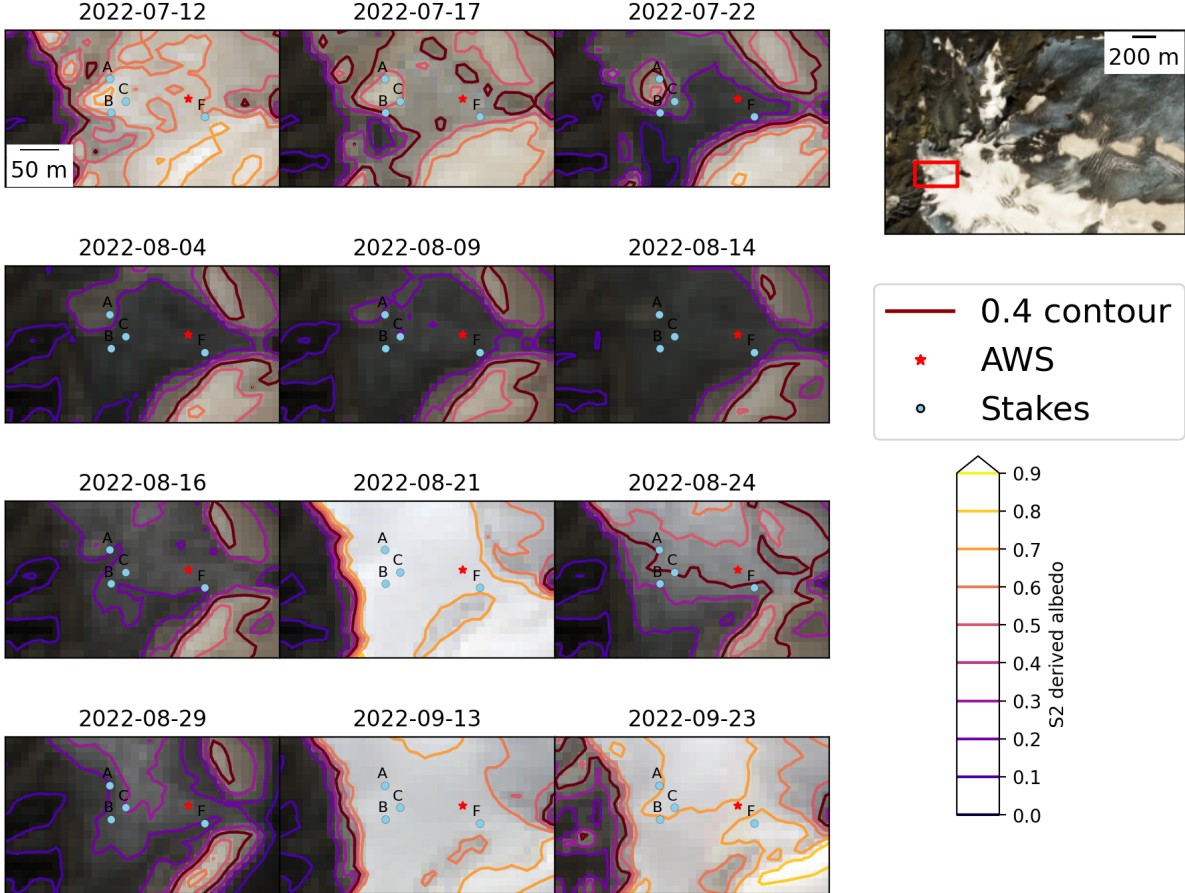

**Figure 8.** RGB composites of cloud free S2 scenes of the Weißseespitze summit region acquired between July 12 and September 23, 2022. The location on the glacier is shown in the inset in the top right (larger crop of the 2022-07-17 scene). The scenes correspond to the point values of broadband albedo as shown in Fig. 7. The positions of the AWS and the stakes listed are marked in red and grey, respectively. The contour lines indicate S2 derived albedo, the 0.40 contour is highlighted in dark red.

### 3.4 Sensitivity of surface mass balance to albedo

#### 3.4.1 Experiment 1: 2022 heat wave

Fig. 9, panel a, shows air temperature and incoming radiation during a week of very warm, sunny weather that preceded the albedo minimum in July 2022 (see also Fig. 7, Fig. 8). Modeled surface mass balance (SMB) is shown for albedo as measured at the AWS and for model runs with a range of constant albedo values (Fig. 9, panels b, c). Daily cycles of SMB during cloud free conditions were predominantly driven by incoming solar radiation. When clouds were present (e.g., July 20 - July 21), the influence of long wave radiation on SMB was more apparent. On July 17 and 18, measured albedo was around 0.40 and modeled daily SMB was about -36 mm w.e. per day On July 22, albedo dropped to "very low" values around 0.20 and daily SMB almost doubled to -71 mm w.e. Considering the entire week-long heatwave, total modeled surface mass loss increased by about 50 mm w.e. if albedo was decreased by 0.1. Total modeled SMB for the week amounted to -302 mm w.e. in the "low" albedo (0.40) scenario and to -407 mm w.e. in the "very low" albedo (0.20) scenario.

Total modeled SMB for albedo as measured at the AWS was -337 mm w.e. during the example period. Surface height change as derived from the SR50 data (filtering process described in section 1 in the supplementary material) during the same time period was approximately -390 mm. Assuming an ice density of 900 kg m$^{-3}$, mass loss during the heatwave period therefore amounts to 351 mm w.e. Camera imagery and S2 data (Fig. 8) suggest minimal remaining snow cover at the AWS during the first days of the heatwave, hence actual mass loss was likely slightly lower. We note that the SR50 data was very noisy during this time and surface height change was strongly smoothed during the data cleaning process, hence a more detailed day-to-day comparison was omitted.

Comparing the energy fluxes during the 2022 heat wave period discussed above (Sec. 3.3 and 3.4.1) to average conditions for 2018 to 2024 during the same time of year highlights that the net shortwave component of the energy balance was the main factor contributing to increased energy available for melt (ME) during the heatwave (Fig. 10). This is due to a combination strong incoming solar radiation during largely cloud free days and "low" to "very low" albedo, which reduced the amount of reflected shortwave radiation compared to average conditions (Fig. 10 panel b). Over the July 15 to 23 period, ME in 2022 was more than double the 2018-2024 average for the same time of year (246% or approximately 18 000 W m$^{-2}$ higher in 2022 than for average 2018-2024 conditions). Reflected shortwave radiation was around 27% (24 000 W m$^{-2}$) lower in 2022 than during average conditions (Fig. 10 panel c). The turbulent fluxes were elevated during the 2022 the heatwave compared to average conditions but their absolute contribution to overall ME is much lower than that of shortwave radiation.

#### 3.4.2 Experiment 2: Average seasonal forcing data and varying albedo

"Low" albedo values and bare ice conditions at the AWS occur between July and September (Fig. 3). Forcing COSIPY with average July-September input generated from the 2018-2024 time series and varying only albedo indicates that the impact of "low" and "very low" albedo is greatest early in this three-month period and decreases as the season progresses and incoming solar radiation decreases (Fig. 11, 12). Average modeled daily surface mass loss in "low" ("very low") albedo conditions was 10 (13) mm w.e. greater in July than during the first half of September. On an average July day on Weißseespitze, bare ice with

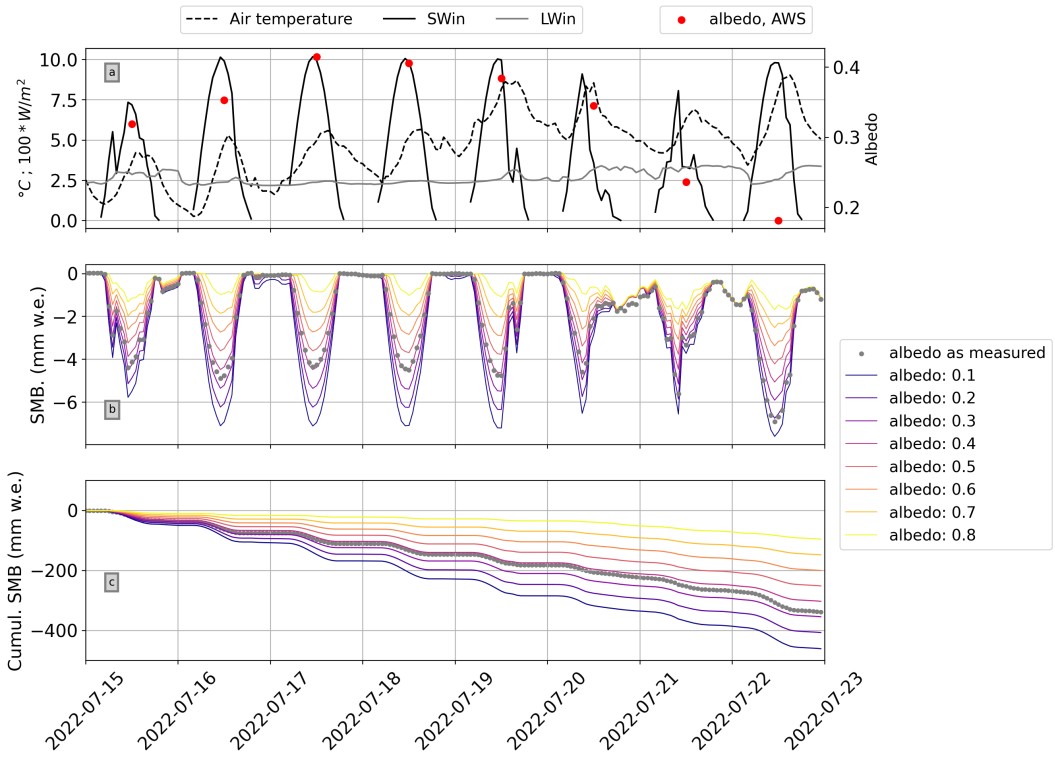

**Figure 9.** a: Incoming short- and longwave radiation, air temperature, and albedo as measured by the AWS during the July 2022 heatwave. b: Modelled surface mass balance (SMB) with albedo input as measured and varied at constant levels. c: Cumulative surface mass balance for the week, albedo input as in panel b.

an albedo of 0.20 was associated with a mean surface mass loss of 38 mm w.e. day$^{-1}$. By late September, the same very low albedo was associated with about 20 mm w.e. day$^{-1}$ of mass loss. The modeled seasonal differences in daily SMB were more pronounced in "very low" albedo scenarios compared to higher albedo. For an albedo of 0.1, the difference between early July and late September was 25 mm w.e. day$^{-1}$. For albedo values of 0.40 and 0.60, the difference decreased to 19 mm w.e. day$^{-1}$ and 13 mm w.e. day$^{-1}$, respectively (Fig. 12).

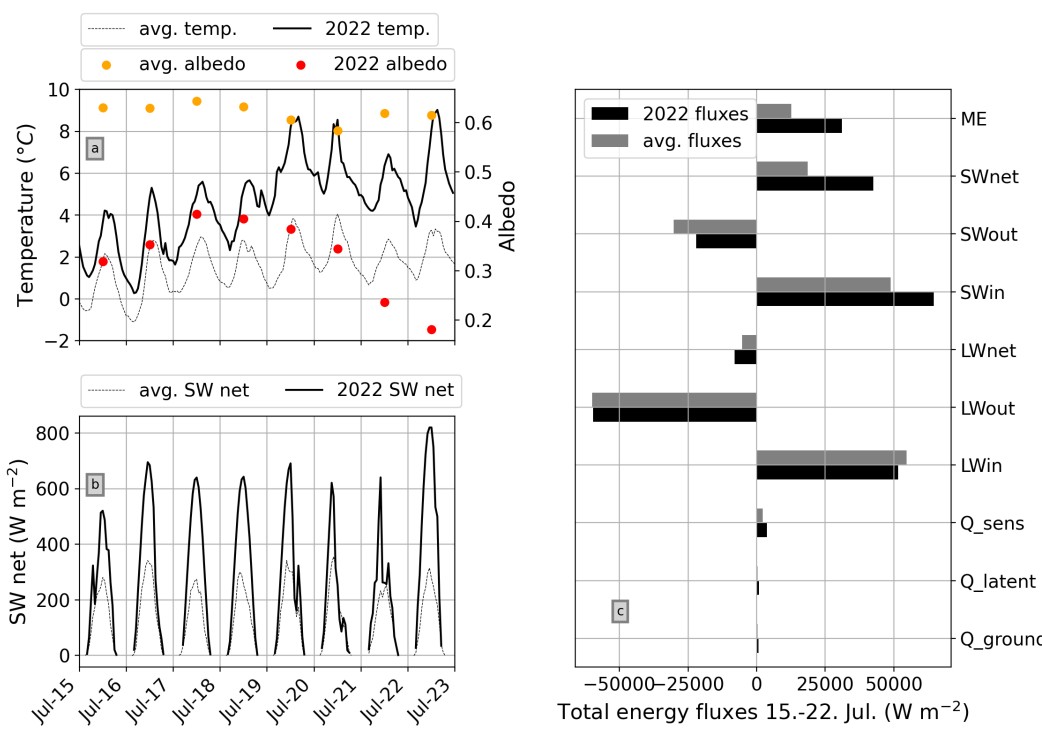

**Figure 10.** a: Temperature and albedo during the 2022 heatwave shown previously in Fig. 7 with average 2018 to 2024 conditions during the same time of year. b: Net shortwave radiation for the same period as in a. c: Modeled energy fluxes for the time period shown in a and b, for 2022 heatwave conditions and average 2018-2024 conditions. ME: Energy available for melt. $Q_{sens}$: Sensible heat flux. $Q_{latent}$: Latent heat flux. $Q_{ground}$: Ground heat flux.

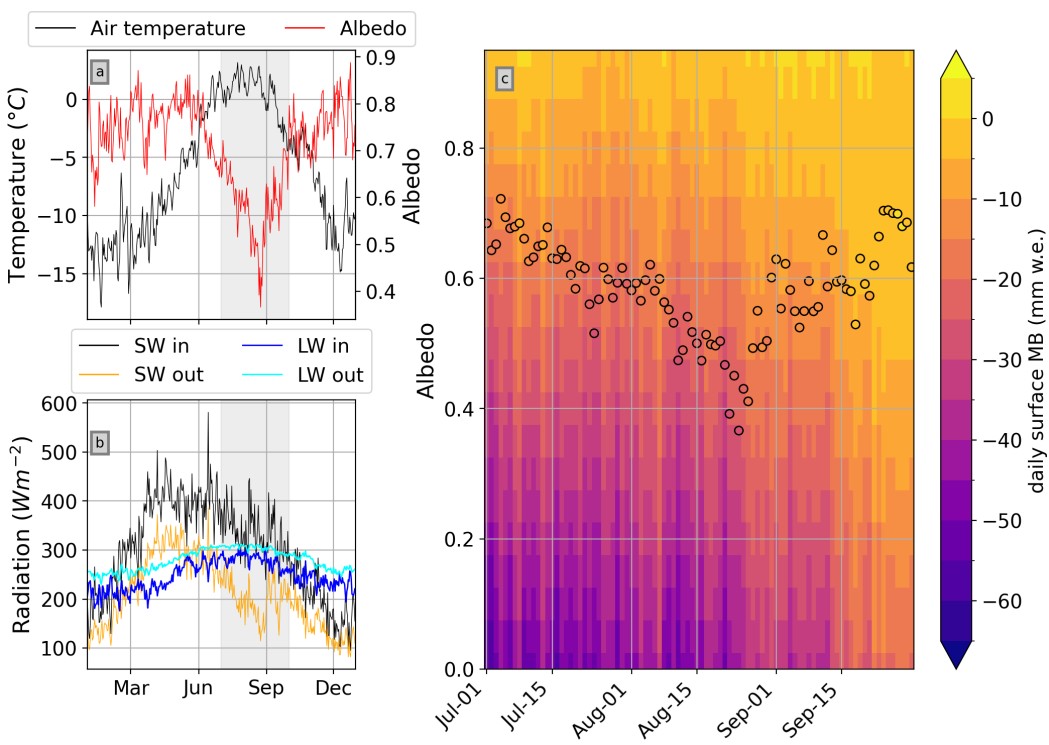

**Figure 11.** a: Daily air temperature and albedo averaged over the 2018-2024 AWS time series. b: Daily mean radiation (short- and longwave, up- and downwelling) averaged over the 2018-2024 AWS time series. Grey shading indicates July 1 to Sep 30, as shown in panel c. c: Modeled daily surface mass balance using the 2018-2024 averages as model input. The circular markers indicate albedo as measured (as in panel a). The background mesh shows modeled mass balance for albedo that is constant in time over the July 1 to Sep. 30 period, varied in increments of 0.05.

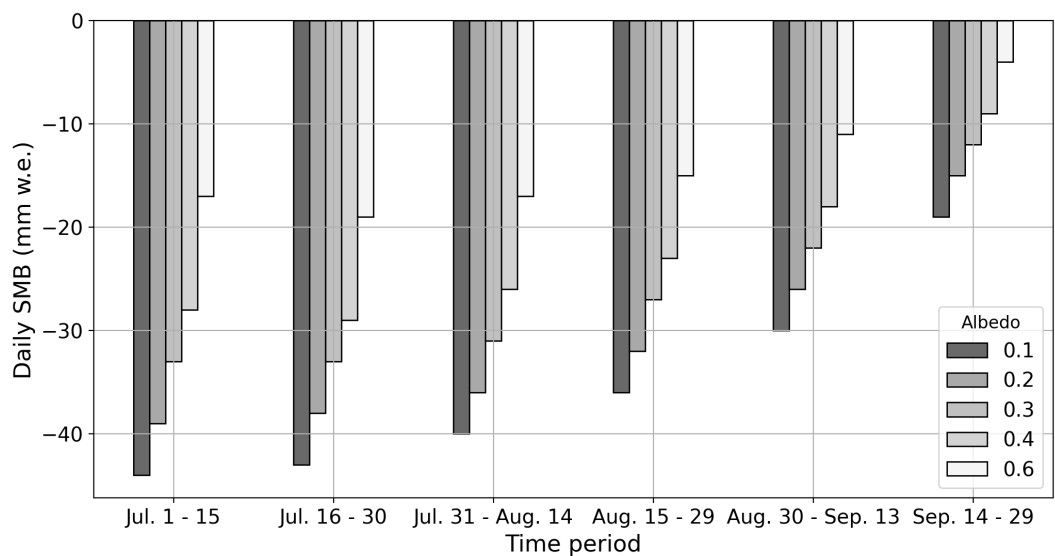

**Figure 12.** Mean modelled daily SMB over 15 day time periods of average July-September conditions and different albedo values. Refer to Table S1 in the supplement for a tabular version of this data.

### 3.5 Interannual elevation dependent albedo variability

To expand the analysis from the summit region of Weißseespitze to the scale of the entire glacier, Fig. 13 shows S2 scenes acquired around the time of the seasonal snow cover minimum for 2018 to 2024 (panels a-f) with corresponding S2-derived albedo binned by elevation bands (panel i). In all years, albedo was very low in the ablation zone up to about 2850 m a.s.l. Interannual variability increased at higher elevations as snow cover in the (former) accumulation zone varied. In the highest regions of the glacier above 3200 m a.s.l., 2022, 2023, and 2024 stood out as darker than previous years and only very limited sections of Gepatschferner remained snow covered.

In the 2022 and 2024 scenes, an extremely dark area was noticeable in the central upper section of the glacier (Fig. 13). The albedo of the ice surface dropped below 0.20 here. An albedo contour line of 0.15 captures the outline of the visually darkest area well and emphasizes the decrease of albedo even beyond the "very low" (<0.2) albedo of the surrounding ice surfaces (Fig. 13, panel h). Similarly dark ice surfaces close to the snow or firn line also occurred in 2018 and, to a lesser extent, in 2023. In histogram visualizations (Fig. 14) of the S2-derived albedo for the scenes shown in Fig. 13, snow cover and bare ice areas are apparent as bi-modal histogram peaks around 0.5-0.6 and 0.20, respectively, from 2018 to 2021. Glacier wide mean albedo ranged from 0.28 in 2018 to 0.41 in 2021. In 2022 and 2024, glacier wide mean albedo dropped to 0.22 and the remaining snow cover did not form a distinct peak, resulting in a uni-modal albedo distribution. 20% and 22% of the glacier area had albedo values below 0.15 in 2022 and 2024, respectively. Most of these very dark areas were concentrated in the relatively high, central section of the glacier mentioned above. 2018 and 2023 similarly had very small secondary peaks, highlighting that a bi-modal albedo distribution separating accumulation and ablation area does not hold in extreme melt years once multi-year firn is depleted. Albedo values for snow free glacier surfaces ranged from 0.10-0.15 to around 0.40, underlining the considerable variability of bare-ice albedo particularly at high elevations (Figs. 13, 14).

We note that the minimum snow cover images coincide with the albedo minima at the AWS within a few days in all years except 2022 and 2023 (Tab. 2). In 2022, the minimum albedo value at the AWS occurred relatively early, on July 22, when there was still remaining snow cover in other parts of the glacier. The image in Fig. 13, panel e, from August 24 shows minimum snow cover conditions at the glacier scale, while a minimal amount of summer snow temporarily brightened the AWS location. In 2023, the albedo minimum at the AWS occurred on August 23, while the minimum snow cover image in Fig. 13, panel f, is from September 10. This is related to cloud cover in the imagery that is closest in time to the AWS minimum. In the days following the AWS minimum, a small summer snow fall briefly brightened the surface. This snow largely melted again but a bit of snow cover remained at the AWS into September, hence albedo at the AWS was relatively high despite minimal snow cover at the glacier scale.

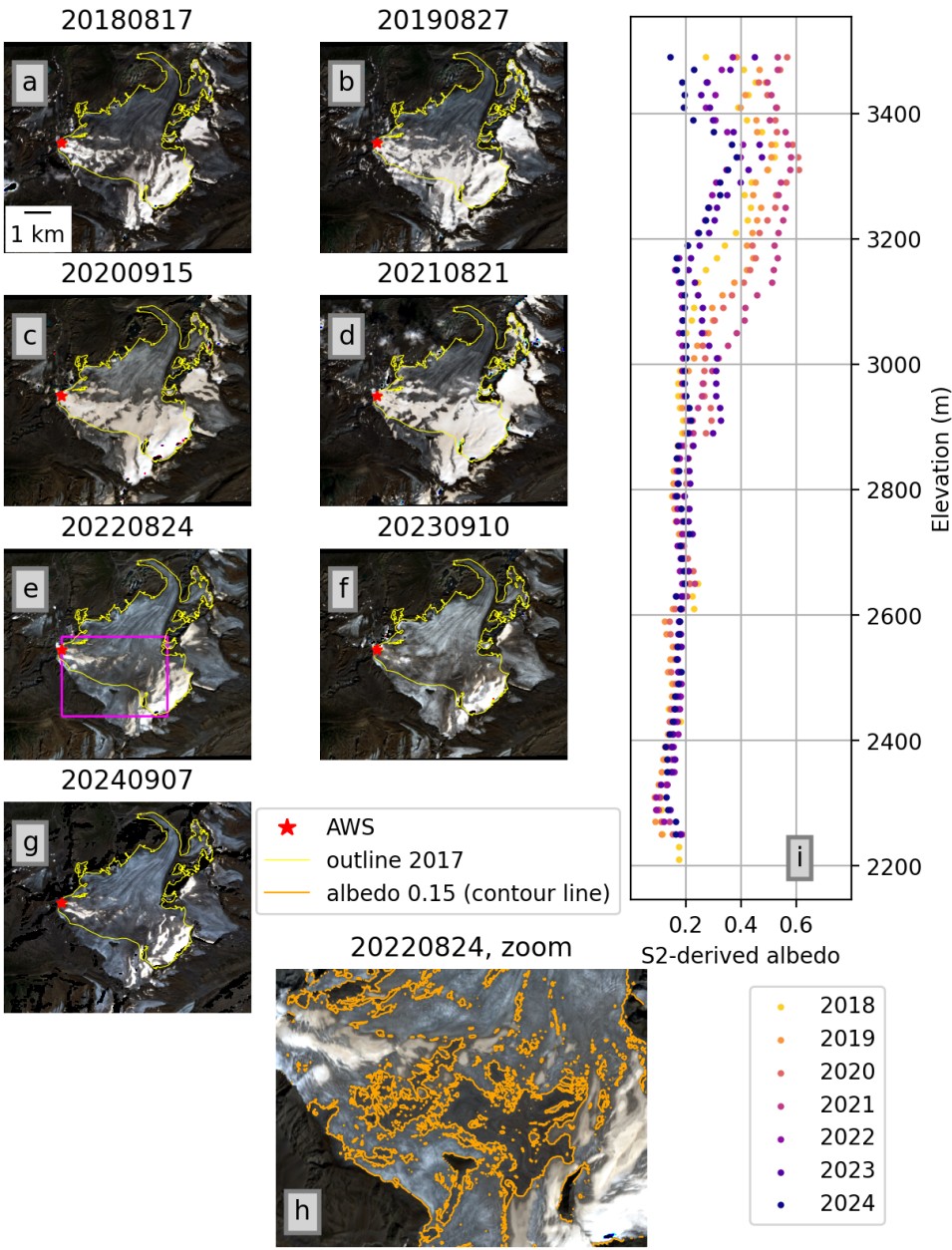

**Figure 13.** a-g: True color composites of Gepatschferner around the time of the seasonal snow cover minimum for 2018-2024. i: Mean albedo per 20 m elevation bands for the same six scenes as in a-f. h: subset of the 2022 scene shown in panel e (magenta box) with contour lines for S2-derived albedo <0.15.

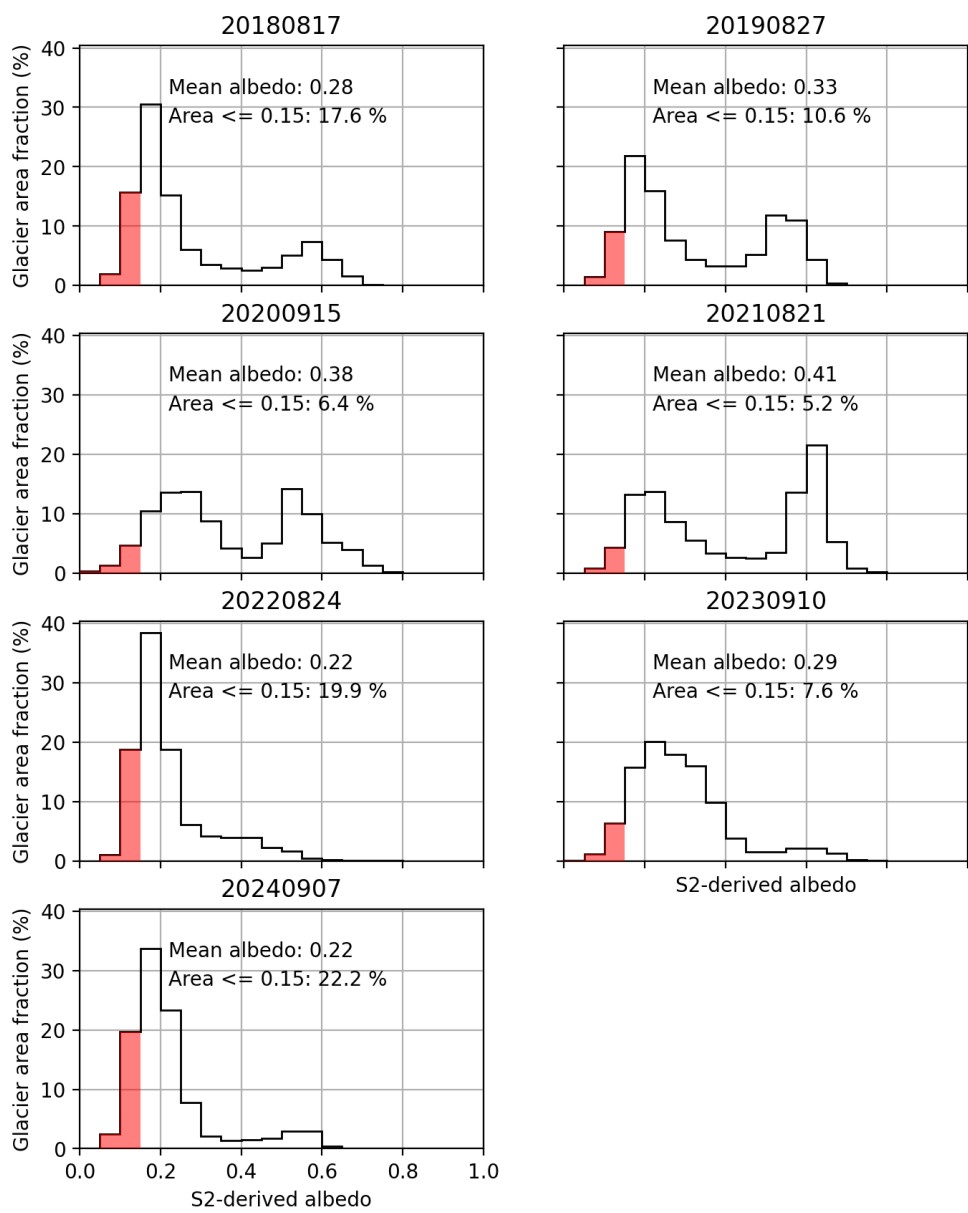

**Figure 14.** Histograms of S2-derived albedo in relation to glacier area for S2 scenes acquired around the time of the seasonal snow cover minimum from 2018-2024, as shown in Fig. 13. Area with albedo below 0.15 is highlighted in red; annotations show glacier-wide mean albedo for each scene.

# 4 Discussion

## 4.1 Method-related limitations

### 4.1.1 S2-derived albedo and comparison with in situ albedo

Generating an approximation of broadband albedo from S2 data over ice surfaces involves a number of challenges, which have been addressed in detail by previous studies. Our approach follows Naegeli et al. (2015, 2017), who found that the narrow-to-broadband conversion of Liang (2001) is suitable for glacier albedo. Naegeli et al. (2017) further showed that the effect of BRDF corrections on ice albedo is negligible. Likewise, the effects of atmospheric corrections have been shown to have only a minor impact on satellite-derived glacier albedo (Fugazza et al., 2016; Traversa et al., 2021). Quantitative uncertainty assessments of surface reflectance products in alignment with GUM standards are complex and subject of ongoing development (e.g. Gorroño et al., 2017; Mittaz et al., 2019; Graf et al., 2023; Gorroño et al., 2024). A strongly simplified estimate based on uncertainty values reported in Gorroño et al. (2024) for a test scene suggests that uncertainties in S2-derived albedo and in situ albedo are of comparable magnitude with 14-16%. However, this estimate neglects uncertainties associated with the narrow-to-broadband conversion itself as well as effects related to unknown errors potentially arising from outdated topographic information (the glacier surface changes constantly, hence digital elevation models used to generate geometrically and atmospherically corrected L2A data do not reflect current conditions), and to scene and surface type specific issues, which have been shown to affect uncertainties in S2 band reflectances and correlations between satellite- and ground measured reflectance data (e.g. Gorroño et al., 2024; Naethe et al., 2024).

In the spatial dimension, the main assumptions underlying our analyses are that the locations of the stakes and AWS in relation to the S2 grid can be determined with enough accuracy to allow a comparison, and that mixed pixel effects are relatively small. The GNSS coordinates (Fig. 2) indicate that the locational uncertainty of the point positions is considerably smaller than the pixel size of the S2 grid. We assume the 5 m buffer applied during the extraction of the S2-derived albedo (section 2.4.1) sufficiently captures the true location of the stakes. We note that the S2 grid is also subject to positional uncertainties. Pandžic et al. (2016) found a potential offset of about 6 m or 0.6 pixels between S2 imagery and an Austrian regional high resolution DEM. The correlation between in situ albedo as measured at the AWS and S2-derived albedo at the AWS position does not change substantially for a 5 m buffer versus a 10 m buffer, hence we consider the 5 m buffer adequate (mean bias 0.076 with 10 m buffer vs. 0.074 with 5 m buffer).

### 4.1.2 Estimated number of low albedo days

We highlight that the estimated number of "low albedo" days as described in section 2.4.1 is indeed an estimate and should be interpreted with caution. The AWS times series allows us to assess how the estimated number of "low albedo" days generated from the S2-derived albedo at the AWS location compares with the observed number (Tab. 4). An underestimation of calculated "low albedo" days compared to the observation is to be expected in cases when albedo drops below the threshold of 0.40 at the AWS and S2 data is not immediately available, or when the S2 time series misses short periods of "low albedo" entirely.

**Table 4.** Estimated number of low albedo days derived via the combined S2 and AWS datasets (Sec. 2.4.1) for the AWS location compared to the number of low albedo days derived from in situ data measured at the AWS.

| Year | S2-derived estimate of low albedo days | Low albedo days in AWS time series |
|------|----------------------------------------|------------------------------------|
| 2018 | 8 | 3 |
| 2019 | 4 | 4 |
| 2020 | 0 | 8 |
| 2021 | 0 | 5 |
| 2022 | 25 | 37 |
| 2023 | 9 | 11 |
| 2024 | 19 | 15 |

In all years except 2018 and 2024, the expected underestimation occured or the estimate of "low albedo" days matched the observation (Tab. 4).

In 2024, the observations did not capture the true number of "low albedo" days due to the prolonged data gap in August. In 2018 there was no such gap, yet the estimated number of "low albedo" days (8) at the AWS was greater than the observed number (3). When comparing ablation with the number of "low albedo" days at the stakes in 2018 (Fig. 6, Tab. 3), stake D also presented a noticeable outlier with 17 "low albedo" days and no ice ablation. In contrast, neighboring stake C had the same number of low albedo days and more than 1 m w.e. y$^{-1}$ of ice loss. This indicates that the linkage between ablation and albedo was not well captured by the S2 data in 2018 for at least some of the stake positions and the AWS. Anecdotally, spatially variable snow depth has often been observed in the WSS summit region during site visits. It seems likely that the 2018 discrepancies at stake D are related to small scale variations in the snow melt pattern that were not resolved in the S2 data.

In 2018, low (<0.40) albedo in the S2-derived time series first occurred on August 17. The albedo observed in situ on the same day was 0.45. The "low albedo" threshold was crossed four days later, on August 21. Observed in situ albedo fluctuated somewhat in the following days, resulting in three days of "low albedo" before a snow fall event brightened the surface on August 25 (Fig. S10, supplement). Taking into account the assumed uncertainties of $\pm14\%$ in the daily in situ albedo (Sec. 2.3) and 16% in the S2 data, it is apparent that the "low albedo" days identified in the S2-derived estimate were within uncertainty of the "low albedo" threshold in the in situ data. This highlights the limitations of categorical threshold choices as well as the considerable uncertainty in the albedo observations.

## 4.2 Albedo and ablation at Weißseespitze

The WSS datasets show the strong albedo impact of snow and firn loss in the upper-most elevations (>3100 m a.s.l.) of Gepatschferner, one of Austria's largest and highest glaciers. Distinct periods of "low" or "very low" albedo are clearly apparent in the in situ and S2-derived albedo. In addition to albedo variability related to snow vs. ice surfaces, bare ice albedo in the

former accumulation zone also varies spatially and from year to year. Ice albedo can reach values as low as those of the surrounding rock in "dark years" (Fig. 8).

All else remaining equal, the difference in ablation between relatively bright bare ice surfaces with albedo values around 0.40 and extremely dark surfaces with albedo below 0.15, as seen in 2022 and 2024 (Figs. 13, 14), is on the order of 10-15 mm w.e. additional daily surface melt for average July-September conditions at WSS (Fig. 12). These sensitivities of SMB to albedo are similar to findings by comparable studies at lower elevations (e.g. Oerlemans et al., 2009; Naegeli and Huss, 2017; Barandun et al., 2022). Naegeli and Huss (2017) showed a decrease of glacier wide mass balance by -0.14 m w.e. yr$^{-1}$ per 0.1 albedo decrease in a study assessing the period 1997 to 2016 for 12 glaciers in Switzerland. They found the greatest influence of albedo changes at the glacier terminus and no impact in the accumulation area, which they noted to be snow-covered year-round and thus not sensitive to bare ice albedo variability (Naegeli and Huss, 2017), in line with findings by Oerlemans et al. (2009). The WSS dataset exemplifies that sensitivity of SMB to albedo change can in fact be very pronounced in (former) accumulation zones as periods of bare ice become longer and occur more frequently. SMB sensitivity to albedo in the summit region of WSS is in the range of Naegeli and Huss (2017)'s results for the lower sections of their study sites.

Besides the absolute albedo of the glacier surface, the amount of time the dark, bare ice is exposed and when in the season this exposure occurs are also key controls of ice ablation on the small summit ice cap of WSS. The observational data show that ice loss at the stakes increases with the amount of low albedo days (Fig. 6). Snow melt patterns in the WSS summit region vary from year to year, which can lead to small scale variability of ablation between the stakes due to the varying length of ice exposure. This snow-driven variability decreased in extreme melt years when the summit region was completely snow free, such as 2022 and 2024. The length of the observed low albedo periods in the time series ranged from 37 days (2022) to 3 (2018). Assuming average August conditions and an albedo of 0.35, the modeled SMB for 37 days amounts to -962 mm w.e. versus -78 mm w.e. for 3 days. This range is broadly aligned with observed ablation for seasons with long and shorter low albedo periods (Tab. 3). Very low albedo was observed at the AWS on 5 to 6 days per year from 2022 onward. 5 days with an albedo of 0.20 result in 31% more modeled ice loss if they occur in late July, like in 2022, rather than early September like in 2024 (-190 mm w.e. vs. -130 mm w.e.). Given the ice depths of 8 to 14 m measured in the summit region in 2018 (Stocker-Waldhuber et al., 2022b), we note that the ablation season of 2022 alone is likely to have caused the loss of around 7 to 12 % of remaining ice thickness.

### 4.3   Open questions: Drivers of very low albedo conditions

The mostly continuous seven-year AWS time series from WSS shows considerable temporal variability in the seasonal albedo minima, with exceptionally low values below 0.20 observed since 2022. At the glacier scale, 2022 and 2024 stand out as "dark" years. This is due to the almost complete loss of snow and firn in these years and a high fraction of bare ice surfaces with "very low" albedo (Fig. 14). Possible explanations for the darkened ice surface in these particular years include the presence of organic and inorganic impurities and stagnant melt water (e.g. Oerlemans et al., 2009; Di Mauro et al., 2017; Di Mauro, 2020; Di Mauro et al., 2020; Barandun et al., 2022; Gilardoni et al., 2022), and may additionally be related to the hydrology and state of the weathering crust (e.g. Takeuchi et al., 2001; Cook et al., 2016; Tedstone et al., 2020; Traversa and Di Mauro,

2024). Reduced nighttime refreezing of supraglacial meltwater during heatwave conditions (e.g., in summer 2022) may have contributed to the very low albedo values in the summit region. The terrain in the upper section of Gepatschferner where summer minimum conditions in 2022 and 2024 were extremely dark (Fig. 13) forms a slight depression where meltwater can pool until new channels are formed that allow it to drain. Based on the S2 imagery, it appears that the dark areas in the upper regions of the glacier were related to impurities that remained and accumulated on the ice surface after snow, firn, or ice layers

melted, and to the presence of liquid water. This process has previously been observed in analyses of an ice core drilled in the accumulation zone of the Adamello glacier (Garzonio et al., 2018). In addition to the presence of water as such, organic or inorganic impurities on the glacier surface appear darker if wet, affecting surface albedo.

Ice cores from the Weißseespitze summit ice cap show distinct peaks of micro-charcoal in near-surface layers (Spagnesi et al., 2023). In a core drilled in 2021, the upper-most 50 cm of ice in particular have high micro-charcoal concentrations (Fig.3

in Spagnesi et al., 2023). Given the ablation records from the stakes, it is likely that this layer partially or completely melted in 2022, exposing ice with higher impurity content and/or causing impurities that were contained in the ice to accumulate on the surface, thus potentially darkening the ice. At present, the linkage of surface albedo and impurities in the cores is intriguing but speculative, pointing to the limitations of broadband albedo for detailed surface type characterization. Resolving the spectral signature of the ice surface via remote sensing imaging spectroscopy or in situ spectroradiometric measurements

in combination with targeted sampling of surface layer ice could provide further insights.

This relates to the need for fine-grained surface classifications at the glacier scale that account for the spatial variability in bare ice albedo, as mentioned above. Interdisciplinary studies combining in situ observations and remote sensing at increased spectral and spatial resolution with modeling are needed to improve understanding of the drivers of bare ice variability (e.g., Zhang et al., 2018; Gilardoni et al., 2022; Barandun et al., 2022; Bonilla et al., 2023).

## 4.4  What does the future hold?

Temperature index based mass balance modeling suggests that Gepatschferner will lose about 95% of its 2017 volume by the end of the century in a future scenario where warming levels are limited to 1.5°C above pre-industrial temperatures; in a 2°C scenario, practically all ice is lost before 2100 (Hartl et al., 2025). In the projections, ice is retained longest in the uppermost regions of the glacier (Fig. 8 in Hartl et al., 2025), including the area of very low albedo in the former accumulation zone

discussed above (Fig. 13).

As firn loss progresses, albedo sensitivity and variability in bare ice albedo will become increasingly important at the highest elevations of Alpine glaciers. At Gepatschferner, multi-year firn has effectively disappeared and conditions as described for the extreme summer of 2003 (Paul et al., 2005) have occurred annually since 2022. Imagining future summer seasons on Weißseespitze and Gepatschferner based on the 2018 to 2024 observations, it is likely that annual mass balance gradients and

glacier-wide as well as distributed albedo will be defined by seasonal snow cover and short term brightening of the upper regions by summer snow falls rather than by the bi-modal divide between the ablation and accumulation zones that was previously a key characteristic of the glacier system. Based on snow height data from the AWS, it is apparent that a deep spring snow pack is not a reliable predictor of a low melt season. Summers with few "low albedo" days, such as 2020 and 2021, were

not preceded by exceptionally snowy winters (Fig. S1). Strong melt years with long periods of exposed ice have occurred in seasons with relatively little snow in late spring (2022) and in seasons with above average spring snow depths (2024). A deep spring snow pack undoubtedly delays the onset of ice ablation and thereby reduces annual ice loss, but recent years indicate that it is not enough to prevent substantial ice loss given otherwise unfavorable conditions. Future years may see more winter snow or cooler summers than in the past three years, but Gepatschferner no longer has a consistent or persistent accumulation zone and hence cannot be expected to survive in the current climate or under future warming conditions (Pelto, 2010, 2011).

In the projected evolution of Gepatschferner mentioned above, the variable characteristics of firn and ice surface conditions and the possible range of bare ice albedo were not taken into account. Considering specifically the very dark zone in the upper section of the glacier, one might imagine a prolonged period of ice exposure of, for example, 20 days in a near-future summer. In our COSIPY experiment for average July through September conditions (Section 3.4.2), modeled SMB for 20 days amounts to -268 mm w.e. for an albedo of 0.6, a typical value for aged snow. For a relatively bright ice surface with an albedo of 0.40, modeled SMB amounts to -438 mm w.e. For very dark ice with an albedo of 0.15, modeled SMB is -655 mm w.e. for 20 days. If the dark ice surface persists in the same area over multiple years, as has been observed at Gepatschferner, the reduced albedo will increasingly affect the spatial patterns of ablation in ways that elevation dependent mass balance gradients do not capture, increasing uncertainties in model projections. Mass balance models that do not resolve albedo variability and the occurrence of very dark ice surfaces may considerably underestimate melt particularly at high elevation in former accumulation zones. Deglaciation may occur sooner than anticipated by current projections if melt-albedo feedback processes are not resolved in modeling efforts. We concur with previous studies that have highlighted the need for improved albedo parametrizations and assimilation of albedo observations in mass balance modeling (e.g. Paul et al., 2005; Naegeli et al., 2015; Naegeli and Huss, 2017; Draeger et al., 2024).

## 5 Conclusions

Ice ablation in the summit region of Weißsseespitze is strongly linked to the duration and timing of bare ice exposure. The length of the seasonal "low" (<0.40) and "very low" (<0.20) albedo periods in the former accumulation zone reached a new record in 2022 with 37 days of ice exposure. Seasonal albedo minima dropped from around 0.3 to below 0.2 in 2022 and did not recover in 2023 or 2024. This suggests increased impurity content at the ice surface since 2022, for example due to impurities previously contained in recently melted firn or ice layers accumulating on the surface, changes to the weathering crust, more supraglacial melt water, or a combination of these factors. Sentinel-2 derived albedo captures the sub-seasonal range of albedo observed in situ well and provides valuable spatial context on seasonal snow melt patterns. Gepatschferner has effectively lost its accumulation zone. Albedo below 0.15 was observed on more than 20% of glacier area in 2024 and the largest very dark sections are found at relatively high elevations, in the former accumulation zone. In future summers, seasonal snow and the spatial variability of bare ice albedo will play a key role for melt patterns. To improve understanding of the driving processes governing ice albedo variability in space and time, multi-disciplinary research that integrates different data and sensor types with modeling approaches is needed. Accurate sub-seasonal and spatially explicit albedo characterizations

are essential for improved projections of future glacier evolution as we shift from glaciers defined by albedo and mass balance gradients aligned with the ablation and accumulation zones to glaciers that no longer retain any multi-annual firn and do not have consistent accumulation zones.

*Code and data availability.*  The stake readings are available on the pangaea data repository:

Stocker-Waldhuber et al. (2025) (https://doi.org/10.1594/PANGAEA.982344)

Meteorological data from the WSS AWS is also available on the pangaea repository and updates will be added in annual intervals to the WSS parent repository: Stocker-Waldhuber et al. (2022a) (https://doi.pangaea.de/10.1594/PANGAEA.939830).

The modified version of COSIPY that accepts measured albedo as an input parameter is available on github as a branch of the main 590 COSIPY repository.

- – Modified version used in this study: https://github.com/baldoa/cosipy_MSc/

- – Main COSIPY repository: https://github.com/cryotools/cosipy

The code to produce the figures and tables in this paper is available at: https://github.com/LeaHartl/WSS_Albedo (a permanent zenodo doi will be generated once revision process is complete.

*Author contributions.* LH, KN, and FC conceptualized the study. MS maintains the AWS and ablation stake network at the study site and contributed to data curation. AB contributed to AWS data curation and energy balance modeling (methodology and software). DF and BD advised on methodology. LH implemented the formal analysis, produced visualizations, and wrote the original draft. All authors reviewed and edited the draft.

*Competing interests.* We have no competing interests to declare.

*Acknowledgements.* L. Hartl is funded by the Austrian Science Fund (FWF) grant number 10.55776/ESP250. A. Baldo and M. Stocker-Waldhuber received funding from FWF grants 10.55776/P34399 and 10.55776/I5246, which also contributed to the installation and maintenance of the instrumentation at WSS. K. Naegeli is supported by the ESA PRODEX Trishna T-SEC project (PEA C4000133711).

We thank two anonymous reviewers and editor Etienne Berthier for their support and time!

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
