# Peer review of "Loss of accumulation zone exposes dark ice and drives increased ablation at Weißseespitze, Austria"

_EGUsphere, 2025_

## Referee Comment (RC2)

**Review of Loss of accumulation zone exposes dark ice and drives increased ablation at Weißseespitze, Austria**

**Overview**
The paper addressed the relevant topic of understanding changes related to glaciers and their accumulation/ablation zones and surface albedo, which can be a challenge to model, especially for future glacier evolution. The study fits well within the scope of TC. The novelty of the paper, although focused on one specific glaciers, is the utilization of remotely sensed albedo and details related to spatial distribution of albedo. The paper is well structured, the figures are clear and easy to follow/read.

I like the way field data and remote sensing is fused and suppose one to another. The introduction reads well and builds the topics well together with ample references. The figures are clean and draw attention to the data. Well made and readable. Much detail to the errors and influencing impacts on observed albedo, both in the paper and the supplemental material, which is very nice to see but perhaps more could have been done to detail the impacts of these errors on the modelled energy balance ? The title of the paper describes well what the paper presents and discusses.

Figures 12 and 13 are very good figures that clearly detail the changes, clean and easy to read. Good work!

Overall the paper is a good read with interesting analytics, clear figures and tells an important story.

**COMMENTS**
The authors might want to review the verb tense throughout the paper, as there are a few instances where it doesn't seem to be used consistently. I'm not an expert in the English language though. A few of such are pointed out in the minor comments.

Overall, I think more could have been done with the Cosipy model since it was setup for the glacier with the unique AWS dataset. This would have provided more insight into various possible changes, details related to the winter mass balance and what would need to happen for the glacier to regain its accumulation area, etc. I think this is one of the shortcomings of the work, which could easily be improved. I also miss some insights to what winter mass balance does for the glacier. Are the low/bare ice albedo years coincidence with winters with little snow ? Could high snow winters help the glacier "back on track" or is it doomed ?

For the References list there are no DOI numbers at any of the references. I assume TC would like some DOI numbers.

Section 2.4 needs some input on the spatial resolution of Sentinel 2 albedo and as well how the buffered pixels are handled. If multiple pixels are covering the buffer what is done ?

Ideally the configuration files for Cosipy and the modified code to account for albedo directly should be included in some repo for reproducibility.

Somewhere in the introduction or in the description of the study site there should be a mention of the winter mass balance, how much it is in context to the melt. Even ballpark numbers if measurements are not available.

In figure 4 there is a very nice comparison of the observed and remotely sensed albedo. For me it seems to be or might be different comparing statistics between seasons or months. Overestimation during the melt seasons and underestimation during winter. Would it be valuable, also in relation to future studies that would like to adopt the methodology, to provide some statistics for this comparison on a monthly basis ? This could be an additional column to table 1.

**MINOR COMMENTS**
L004: feedback or relationship would be better than connection
L007: remove conditions …low albedo values…
L007: Recorded by what ? AWS or S2 ?
L008 Is this only ice ablation? Or just general ablation for the location ? Winter snow + firn + ice?
L016: The new Glambie paper in Nature would be a good additional reference here
L021: More context is needed. For example: "…bare ice becomes exposed with lower surface albedo than firn" This is then referred to in the next sentence.
L026: glacier albedo might be better here. Or snow and ice albedo. Same in L027
L039: Drive is not a good word here. Force or input data
L056: …during extreme years… extreme years of what ? Rewrite this sentence so it is clear what is extreme.
L057: In the following of what? In this study… could be used
L070: For non-native Alps people writin Gepatschferner *glacier* would be helpful
L071: I am a bit confused by this number, 0.05 km2, as the red area highlighted in Fig.1 showing the ice covered mountain seems to be more than ~1km ? Are you referring to the absolute peak of the mountain ? Perhaps, if this number is not important you could skip it to reduce confusion ? Is it possible that this is the area all the stakes are aggregated in fig 1 panel c ? Perhaps a small box there would be useful.
L073: what is cal BP ? Please write out.
L073: change ice depth to ice thickness.
L095: Are they operated only during summer ? Redrilled in each spring. In L096 it is indicated the stakes are visited over the full year or is this inly for the summer. In the ablation stakes section (2.2) it might be clear if these stakes only survey summer mass balance or if the survey winter mass balance as well ?
L111: I would suggest to use SW_in and SW_out which I believe is pretty standard ? Or provide some insight in why *ref is used. Reference or reflected ? On most other locations in the paper SW*in and SWout is used. Please systematically go though the paper and make these the same in text and figures.
L149: In this sentence "Multi-spectral reflectance was converted to broadband
150 reflectance using the conversion developed by Liang (2001), following prior work addressing broadband albedo of glacier surfaces (Naegeli et al., 2017)." A bit more insight into what equation is used would be beneficial and what bands are input.

L174: are => were and in L115

L179: the word model or simulate is missing after the citation, "to **model the** surface mass balance…"

L182: take => use

L186: To Cosipy, is the snowfall in height units or w.eq. ? Is there a conversion with density done ?

L189: Could you explain a bit what idealised runs actually are ? Observed met data with 0.05 incremental changes in albedo to the full period from and to ? Running from fall to fall or during summer ? What is the timestep of the model runs needs to be added.

L191-192: What does this sentence mean ? What assumptions for the subsurface are assumed ? Is something in the model skipped ? I would assume that there are some sort of initial conditions for the model ?

L198: *"However, running the model with input as recorded by the AWS for a sub period with SR50 data of acceptable quality shows good agreement between the modeled and measured surface height (Fig. S8, supplementary material) during the phase of snow melt and subsequent ice ablation at the station."* This sentence would improve by being re-written for clarity, it is a bit . Surface height should be surface height changes.

L198: For me it is a bit too simple to say: *"Detailed validation exercises are beyond the scope of this study."* The authors mention a good comparison to observed and modelled surface height changes which in a sense is a good calibration metric. Since there is a figure in the supplements please add some simple stats in the text of this agreement, i.e. rmse, R2 and bias for example.

L206: Add the years (2018 - 2024) in the text so it is clear that these are mean values for the whole study period.

L211. Are these anomalies ? See comment for figure 3

L246: Cumulative ice ablation. This is summer mass balance right ? No only ice, but ice and snow is it not ? Or skip the word "ice"

L256: is => was

L261: mm w.e => mm w.e. (a dot is missing)

L353: "broadband conversion of Liang (2001) is suitable for ice albedo." Not for snow albedo ? Perhaps expand this a bit so it is clear.

L373: observation => observations

**Figure 1:** In the map (b) of Gepatschferner there are a lot of green outlines. I would suggest highlighting what is defined as Gepatschferner with another color for context and then either skip the other outlines or have them in an alternative color. A lot of ice patches and small glaciers are in the area. Also, there are parts of glaciers that seem to have no outline, i.e, in the south west part of the image. It also seems like parts of the Gepatschferner are missing its outline ?

**Figure 3.** I am a bit confused by the figure. I understand that the long term mean albedo is shown in each subplot but for each individual year are the anomalies shown (data minus the long term mean) or the change compared to the mean ? For example, for the big grey period in 2024 in September is the anomaly about 0.4 lower than the long term mean or is the observed albedo about 0.2 ? I would change this figure to true anomalies (data minus mean). That would also highlight better the

deviations as positive or negative. The AWS and S2 validation/comparison is already done in Figure 4 and 5 so there is no need to repeat that here with the black dots.

**Figure 7:** In the b-panel the color needs to be different for SWin and SWout. It is very hard to see what is what. Perhaps change to SWnet ? Or have SWin and SWout with different sign as often is done?
**Figure 8** is a very nice figure.

Figure 10: It would be a good idea to have the same names for radiation components as mentioned earlier, SW/LW in/out here but different earlier in the text.

Figure 11: Meam should be mean

---

## Author Comment (AC1)

RC1:

Thank you very much for the encouraging comments! We truly appreciate the constructive feedback and the time spent on the review! Please see below for responses to specific comments (our responses are in blue text).

-Lea Hartl on behalf of the authors

The study focuses on quantifying the evolution of albedo in the accumulation zone of the WSS (Austria). By combining in situ measurements with satellite observations, the authors demonstrate a good agreement between these approaches, enabling a detailed analysis of the spatiotemporal variability of albedo. On this glacier, as on many other Alpine glaciers, accumulation zones are progressively shrinking, exposing not only firn from previous years but also an increasing proportion of exposed bare ice at the surface, leading to a significant decrease in albedo. Based on seven years of observations, the authors highlight a pronounced decline in albedo within the accumulation zone, where ice exposure at the surface has become more frequent, particularly since 2022. Due to the positive feedback of albedo, this phenomenon further enhances glacier melt, a process the authors quantified in this study. Despite its critical role in glacier mass balance evolution, this mechanism remains relatively understudied, particularly in terms of its spatial variability. Results presented in this study thus provide valuable insights for the glaciological community.
I thoroughly enjoyed reading this paper, which is clear, well-written, and well-structured, presenting novel and valuable findings. I particularly appreciated the discussion section, especially sections 4.3 and 4.4. The introduction is also well-written and clear; however, the list of cited references is often not exhaustive. It would be beneficial to either expand the references or, at a minimum, indicate their non-exhaustive nature using "e.g." (e.g., lines 1, 20, 26, 31, 36...). The methodology section is well-structured and engaging, but data availability (e.g., time period, resolution, etc.) remains somewhat unclear at times. Given that numerous measurements and observations were conducted over different periods and at varying temporal and spatial resolutions, a more visual representation (such as a figure or table) summarizing the measurement periods, resolution, uncertainties, etc., could significantly improve readability. Finally, before publication, I believe it would be valuable to clarify the (not major) points outlined below.

We will address the points mentioned above (expand reference list, indicate that it is not exhaustive) and add a data overview figure (example figure showing data type and

[Figure]

Incertitudes

It seems important to better clarify and quantify uncertainties, particularly to improve the discussion and comparison between the two methods.

1) Regarding the AWS observations, I appreciate the effort to quantify uncertainty presented in the appendix, but some questions remain: What is the sensor accuracy (line 137)?

The uncertainty in daily albedo due to sensor accuracy is 14%, derived from the approximate uncertainties for daily values at mid-latitudes given in the sensor manual. We added this to the revised manuscript in parenthesis. The sensitivity of the sensors is 12.81x10^-6 and 12.83x10^-6 V/ (W/m2) for the up and down facing sensor, respectively. While this is an important metrological metric, the error computation that follows from this and other factors is complex and we therefore follow the guidance of the manual.

Where does the 14% uncertainty come from (e.g., Figure 4 and S5)?

Section 2.3. states: "Based on standard error propagation (root sum of the squares) for the ratio of up- and downwelling radiation, we assume an uncertainty of 14% for the daily albedo". The up- and downfacing sensors each have an uncertainty of 10%. 10^2 + 10^2 = 200. The root of 200 is 14.1.

The uncertainty related to surface roughness could be mentioned (e.g. line 131)

Added a note on this in the suggested line with additional references.

2) The S2 images provide high-resolution spatially distributed albedo data, which is highly relevant. However, uncertainties associated with this method are not quantified and should be reported in the study (e.g., Figure 4). Additionally, for this method, it is unclear whether and how the solar zenith angle at the time of image acquisition is accounted for in the albedo calculation. Could this have a significant impact on the albedo from S2?

Thus, the comparison between AWS and S2 approaches (e.g., bias, RMSE, Section 3.1.2) could be further discussed in relation to their respective uncertainties.

Quantifying uncertainties in S2 and similar earth observation data is highly complex and subject of ongoing work in the EO and metrology community (e.g. Mittaz et al 2019). Challenges arise from the numerous sources of uncertainty and processing steps that take place between the collection of raw at-sensor telemetry (L0) and derivation of calibrated top of atmosphere radiance (L1), and bottom of atmosphere reflectance (L2), which involve radiometric, geometric, and atmospheric corrections, and principles of radiative transfer. We refer to Gorroño et al (2017, 2024) for an overview of uncertainty contributors combined in accordance with the GUM. The uncertainty related to solar zenith angle is assumed to be negligible compared to other sources of uncertainty by Gorroño et al (2024). Gorroño et al (2017) provide further context on this. We note that at present there is no universal, standard way of quantifying or reporting uncertainties in L2 products. We agree with the reviewer that this would certainly be desirable and that it is beneficial to attempt rough uncertainty estimates even though exact quantification is beyond the scope. We propose an approach based on values reported in Gorroño et al (2024) for the S2 L2A product per spectral band (Table 2 in Gorroño et al, 2024). We use the values generated with the MCM approach for the Amazon scene. For the bands used in our albedo computation, these are:

B2: 9.75%; B4: 7.77%; B8: 1.12%; B11: 1.89%; B12: 3.30%

The sum of the squares for these values is 171.15 and the root thereof is 13. Accounting for the correlation between the bands (Gorroño et al, 2024, Fig 5) adds 3%, resulting in an estimated uncertainty of 16% We have included this uncertainty estimate in Fig. 4 and expanded the discussion of this issue (including comments on assumptions made in this estimate) in the revised manuscript.

Gorroño, J., Fomferra, N., Peters, M., Gascon, F., Underwood, C. I., Fox, N. P., Kirches, G., & Brockmann, C. (2017). A Radiometric Uncertainty Tool for the Sentinel 2 Mission. *Remote Sensing*, *9*(2), 178. https://doi.org/10.3390/rs9020178

J. Gorroño, L. Guanter, L. Valentin Graf and F. Gascon, "A Framework for the Estimation of Uncertainties and Spectral Error Correlation in Sentinel-2 Level-2A Data Products," in *IEEE Transactions on Geoscience and Remote Sensing*, vol. 62, pp. 1-13, 2024, Art no. 5634613, doi: 10.1109/TGRS.2024.3435021.

Mittaz, J., Merchant, C. J., & Woolliams, E. R. (2019). Applying principles of metrology to historical Earth observations from satellites. *Metrologia*, *56*(3), 032002.

Spatial variability and albedo

1) The study of the spatial variability of albedo is both interesting and innovative and could be better illustrated. For example, lines 243–244 and Section 3.2 could be accompanied by a spatially distributed albedo map, as this information is not clearly visible in Figure 5.

Figures 8 and 12 show spatially distributed albedo as contour plots. Section 3.2 addresses time series of albedo at point locations, so a single albedo map would not convey the same information. We are happy to include additional maps of spatially distributed albedo if that would be beneficial but would ask for a clarification of what sort of time frame should be considered in maps that would accompany section 3.2.

2) The interpretation of the relationship between albedo and SMB is valuable; however, the effect of temperature is not discussed. Years with low albedo values (e.g., 2022–2024) are also among the warmest, making it difficult to disentangle the impact of albedo feedback on melt from that of high temperatures. This aspect could be further explored or at least mentioned. Additionally, temperature data are reported in Figure 7b, but this panel is not discussed when analyzing the summer of 2022.

Section 3.3 on summer 2022 does include comments on the temperature data shown in Fig 7b. We have included a reference to panel b in the text to make this more clear. We also added a paragraph and an additional figure to section 3.4.1 showing the energy fluxes during the 2022 heatwave compared to average 2018-2024 conditions. The net shortwave component of the energy balance was the main factor contributing to increased energy available for melt (ME) during the heatwave. The turbulent fluxes (which are related to air temperature) were elevated during the heatwave compared to average conditions but their contribution to overall ME is much lower than that of shortwave radiation (as can be expected for mid-latitude glaciers in summer). We quantify the contributions in the revised manuscript.

[Figure]

*Additional figure showing the 2022 heat wave and average 2018 to 2024 conditions during the same time of year and associated energy fluxes in panel c: ME: Energy available for melt. Q_sense: Sensible heat flux. Q_latent: Latent heat flux. Q_ground: Ground heat flux.*

3) Furthermore, regarding the results presented in Figure 12, it seems important to mention that the satellite images for 2023 and 2024 were acquired in September, a period when the glacier's albedo is likely not at its lowest (see Figure 2). Could the comparison of glacier-wide albedo between years be somewhat biased by the late acquisition dates in these two years (e.g., line 343)? This point could be addressed.

The images were selected for minimum snow cover and all suitable (cloud free) images were considered. Both 2023 and 2024 had relatively long ablation seasons that extended into October. In 2024, the lowest albedo at the AWS was measured on September 8 and the S2 image in Fig 12 is from Sep. 7 (see Fig 3 and Table 2). The image and the AWS minimum coincide very closely. In 2023, the lowest albedo value at the AWS occurred on August 23. A small snowfall event then brightened the surface. This snow melted again in the following days across most of the glacier, although some snow remained at the AWS location. The S2 image from August 24 is partially affected by clouds and cloud shadows (left image in the figure below). In the S2 image from September 10 (shown in Fig. 12 and

below on the right) the remaining seasonal snow cover has retreated further compared to the Aug. 24 image, although a minimal amount of snow from the summer snowfall in late August remains at the AWS location. The exposed bare ice surface appears visually brighter in some areas of the Sep. 10 image but we do not believe this indicates a bias due to the time of year.

[Figure]

Simulations

Although the authors mention that the model validation exercise seems to be outside the scope of the study, the fact that they quantify the impact of albedo on melt using this method and highlight it in the results (e.g., lines 324-327, Figure 11), and also in the discussion, conclusion, and abstract makes it, in my opinion, difficult to avoid an precise evaluation of the model. If this precise quantification is a result the authors wish to emphasize, I strongly encourage them to properly evaluate the model. Otherwise, I suggest they focus solely on relative comparisons from sensitivity tests.

Thank you for this feedback. Reviewer 2 had similar comments and we have expanded and restructured the section explaining the sensitivity experiments to provide more information and quantitative comparisons of modeled and observed surface height change. The supplementary material pertaining to this has also been expanded. Possibilities for model evaluation are limited by the available observational data, mainly the surface height change information derived from the SR50 records. There is a lot of noise in this data set and small changes in particular cannot always be reliably extracted. We have explained the limitations of the SR50 in more detail and provide comparative statistics on modeled and observed surface height change for an example period with relatively good data quality in the revised manuscript. We would like to keep the absolute melt values from the different model runs as part of the results but will reduce focus on these towards more frequent usage of relative values in the revised manuscript.

Some suggestions for model evaluation : The simulated vs. observed snow-to-ice transition could be quantified using a delta day (line 201) ; Fig 9 could be evaluated using the SR50 (although the SR50 was used to force the model with snowfall, there is no precipitation during the period presented); ice temperature sensor measurements could be compared with the model simulations.

We have added a delta day quantification and additional statistics for the evaluation period detailed in section 2.5. We also provide a quantitative comparison of the cumulative modeled and observed surface height change for the period shown in Fig. 9. The quality of the SR50 data is not ideal during this period, which we also explain in this section. The "model evaluation" figure in the supplement has been adapted to show a comparison of daily surface height change.

[Figure]

*Updated supplementary figure showing measured and modeled surface height change, and observed albedo and snowfall during the evaluation period (upper panel). The lower panel shows modeled and observed daily change rates.*

Additionally, some clarifications on the simulation parameterizations seem important, particularly regarding model calibration and initial conditions (e.g., in the appendix). Finally, is the surface in the simulations in Figure 11 still ice? This should be specified. If so, is an albedo of 0.6 realistic for ice?

Following this comment and similar points by reviewer 2 we have restructured section 2.5

(sensitivity experiments) to more clearly explain what was done. Regarding Fig. 11: we assume an ice surface covered by a minimal amount of snow, like would be the case after a small summer snowfall. This is now clarified in Sec. 2.5, along with the main assumptions made regarding initial conditions. We will prepare a code repo with the relevant config files to accompany the revised manuscript.

**Line by line comments:**

Lines 27-31: long sentence difficult to understand. Please reformulate.

Split into multiple sentences as follows: "At local and regional scales, ice albedo depends on meteorological factors like solar elevation, cloudiness, and radiation budget (Volery et al., 2025) and on surface roughness (Irvine-Fynn et al., 2014). Additionally, the presence of liquid water on the ice surface and the characteristics of the pore space and the weathering crust impact albedo (e.g. Dadic et al., 2013; Traversa and Di Mauro, 2024). Light absorbing impurities of organic and inorganic origin, including carbon, algae, and dust, can lead to albedo reductions and darker glacier surfaces (e.g. Oerlemans et al., 2009; Di Mauro et al., 2017; Goelles and Bøggild, 2017)."

Line 80 to 88: What is the measurement period for the AWS (in relation to the above comment) and other sensors, and what is the temporal resolution? The same applies for the ice temperature sensors and the camera.

The measurement period for the AWS (including all associated sensors) is from the date of installation to present with some minor gaps due to power supply failures and other technical issues. Data are logged as ten minute averages, as stated. The measurement period for the camera is also from the stated date of installation to present. The camera takes a picture every two hours during daylight hours. We have added a note to this section clarifying this.

Line 83: Ice temperature sensor: Is it used in this study?

No. The information from the thermistor strings informs the initial ice temperature assumptions made in the energy balance model in a general way but this sensor is not essential for the study. We mention it here for completeness along with the other components of the AWS. We can remove this if it is confusing or seems unnecessary.

Line 85-86: It took me some time to understand that we are in the accumulation zone, but with exposed ice (which is uncommon for an accumulation area). A clearer description here could help, especially since Figure 1 shows only snow.

Added the following description and changed figure 1 to show an image with exposed ice. "The location of the AWS is in the highest region of the glacier within the (former)

accumulation zone. Multi-year firn is no longer present around the AWS and bare ice is exposed if the seasonal snow cover does not persist through the summer."

Line 105: "Ice flow is not apparent" – Please specify: "Ice flow is lower than..."

We would prefer to keep the current phrasing. We do not have a precise way to quantify "lower than". In the GNSS coordinates gathered during site visits, no systematic shift of the position of the AWS or stakes over time is apparent. The coordinates form a "cloud" without an apparent direction of movement. Hence, any movement due to ice flow would be less than the locational uncertainty of the points. The image below shows the GNSS coordinates of the AWS as recorded during visits.

[Figure]

Line 139 and throughout the document: "Low" and "very low" refer to specific values (i.e., 0.2 and 0.4) as indicated here. These terms are used throughout the document, sometimes with quotation marks and sometimes without. Conversely, "low albedo" is sometimes mentioned without explicitly referring to these values, making the text harder to follow. Please ensure that quotation marks are consistently used when referring to these specific values, or alternatively, use a uniform notation (e.g., alb < 0.2).

We will go through the manuscript to ensure consistency.

Line 145: Could you provide further details, such as the spatial resolution of the S2 images, the number of images, and the period covered?

We have added the spatial resolution (10x10m) and clarified that we use S2 data for the 2018-2024 period, i.e. the period of record of the AWS and stake data. In section 2.4.2 we additionally specified that we select one image per year to assess minimum snow cover conditions.

Line 182: "Albedo as input" – do you mean albedo from the AWS?

In this case yes, although any kind of albedo data could be used. We specify this in the revised manuscript.

Line 216: Unclear where the value of 0.3 comes from. Could you clarify?

This refers to data as measured by the AWS. We have added references to the respective table and figure. We also removed unfortunate typos in the table, which likely caused this confusion (sorry).

Line 234, 238: "Generally coincide or occur" – This statement could be quantified (e.g., using delta day) to add more weight to the comparison.

We added a note on this in the revised manuscript. In 2018, S2-derived "low albedo" was observed five days prior to "low albedo" conditions at the AWS. Otherwise the S2-derived low albedo periods are within the AWS low albedo periods. In line 238, the statement is followed by a description of cases when there are discrepancies with examples. The stakes are not expected to have low albedo at exactly the same dates due to varying snow melt patterns. The irregular nature of the S2 time series makes it challenging to meaningfully interpret shifts of a few days - these may be due to snow melting earlier or later at one stake compared to the next, or to differing availability of S2 imagery.

Line 248 and throughout the paper: Be consistent with units: mm w.e. should be accompanied by a time period (e.g., mm w.e. yr$^{-1}$) (e.g., line 248, line 323, line 396, etc.).

Added time periods where appropriate. In cases where it is not a year or a day, the period is stated in the text (e.g. "X mm w.e. over five days").

Line 246 to 251: To give more weight to the different ablation values, you could mention the percentage they represent relative to the mean, especially for 2022.

We would prefer to keep the focus on absolute values rather than percentages of a mean. The stake data exhibit high year-to-year variability, which would not be well represented by averaging over all years. The large variability is - in our opinion - the key characteristic of the stake data set from this location and more relevant for the future development of the glacier than the deviation of single years from an average value.

Line 252: Specify that this refers to albedo from S2.

Added this clarification.

Line 419: While I find this discussion relevant and convincing, wouldn't the primary effect of darkening at the glacier scale be more related to the expansion of the accumulation zone?

The darkening is due to both the loss of snow/firn area and unusually dark ice surfaces in these years. We clarified this in the revised text and added a reference to the figure showing albedo histograms: "At the glacier scale, 2022 and 2024 stand out as very low albedo years. This is due to the almost complete loss of snow and firn in these years and a high fraction of bare ice surfaces with "very low" albedo (Fig. 14). Possible explanations for the darkened ice surface in these particular years include….."

Line 459-460: Also, the firn.

Changed to "firn and ice surface conditions"

**Figures and Sup. Mat.**

General: Many figures should be larger because they are sometimes difficult to read.

We have adjusted font sizes and figure design to improve this.

Figure 1b: Glacier outlines and 50m contours are barely visible (green and blue lines).

Changed the color and increased the linewidth.

Figure 5: Difficult to read. Consider splitting it into two panels: one with AWS-in situ and AWS-S2 (to show the comparison) and another with S2 at the stakes (to show spatial variability).

We have split the figure as suggested.

Figure 7: It is difficult to differentiate the colors specific to each stake. Add (a) and (b) directly on the figure.

Panel labels (a, b) are included in the figure. We have adjusted the colors to hopefully make them easier to differentiate.

Figure 12h: The contours are hard to see, especially the green ones. Could they be made larger or changed to a different color?

We changed the color and kept only one set of contours to make the figure cleaner and easier to interpret.

Sup. Mat.: The order of references to the supplementary material is not always chronological (e.g., Line 90: referenced as 2 in the supplementary material but should be 1). Additionally, some references to the supplementary material could be more specific (e.g., Line 186: which figure does this correspond to?).

We revised the structure of the supplement to be in chronological order and have added more specific references to the supplement in the main manuscript.

---

## Author Comment (AC2)

RC2:
Thank you for the helpful feedback and comments and the time spent on the review! We will address the points raised to the best of our abilities in the revised manuscript. Please see below for responses to specific comments (blue text).
-Lea Hartl on behalf of the authors

Review of Loss of accumulation zone exposes dark ice and drives increased ablation at Weißseespitze, Austria

Overview

The paper addressed the relevant topic of understanding changes related to glaciers and their accumulation/ablation zones and surface albedo, which can be a challenge to model, especially for future glacier evolution. The study fits well within the scope of TC. The novelty of the paper, although focused on one specific glaciers, is the utilization of remotely sensed albedo and details related to spatial distribution of albedo. The paper is well structured, the figures are clear and easy to follow/read.

I like the way field data and remote sensing is fused and suppose one to another. The introduction reads well and builds the topics well together with ample references. The figures are clean and draw attention to the data. Well made and readable. Much detail to the errors and influencing impacts on observed albedo, both in the paper and the supplemental material, which is very nice to see but perhaps more could have been done to detail the impacts of these errors on the modelled energy balance ? The title of the paper describes well what the paper presents and discusses.

Figures 12 and 13 are very good figures that clearly detail the changes, clean and easy to read.
Good work! Overall the paper is a good read with interesting analytics, clear figures and tells an important story.
Thank you!

COMMENTS
The authors might want to review the verb tense throughout the paper, as there are a few instances where it doesn't seem to be used consistently. I'm not an expert in the English language though. A few of such are pointed out in the minor comments.
We have revised the manuscript to improve the consistency of tenses. We aim to generally use past tense for the methods and results and present tense for time invariant content and interpretations. We are also not experts in the English language and are happy to correct instances we may have missed during this round of revisions.

Overall, I think more could have been done with the Cosipy model since it was setup for the glacier with the unique AWS dataset. This would have provided more insight into various possible changes, details related to the winter mass balance and what would need to

happen for the glacier to regain its accumulation area, etc. I think this is one of the shortcomings of the work, which could easily be improved.

Based on this and comments by R1, we added a figure to the results section partitioning the fluxes for average 2018-2024 summer conditions and the 2022 heatwave. This shows that shortwave radiation is a key driver of energy balance at this site.

The focus of this study is the relationship between summertime melt and the duration of ice exposure and ice albedo at high elevations, hence the COSIPY experiments target this aspect. The snow height data from the AWS is very noisy (as detailed in the supplement) and we do not have snow density measurements, hence it is not possible to fully quantify winter mass balance. In the revised manuscript, we have added general information on the relationship between winter and annual mass balance to the study site description (see also responses to below comments), and expanded the discussion of this issue.

[Figure]

*Additional figure showing the 2022 heat wave and average 2018 to 2024 conditions during the same time of year and associated energy fluxes in panel c: ME: Energy available for melt. Q_sense: Sensible heat flux. Q_latent: Latent heat flux. Q_ground: Ground heat flux.*

I also miss some insights to what winter mass balance does for the glacier. Are the low/bare ice albedo years coincidence with winters with little snow ? Could high snow winters help the glacier "back on track" or is it doomed ?

There are no winter mass balance data for this site. Two WGMS reference glaciers very close by do have seasonal records. Based on those time series, it is apparent that the trend in ablation (summer mass balance) drives the annual trend in this region. The below figure shows winter and summer mass balance vs. annual mass balance for the two closest WGMS reference glaciers, Hintereisferner and Vernagtferner, to illustrate this. We have added commentary on the questions posed by the reviewer in the discussion section (sec. 4.4.) and added a figure to the supplement showing the time series of 2018-2024 snow depth. Summers with few "low albedo" days, such as 2020 and 2021, were not preceded by exceptionally snowy winters. Strong melt years with long periods of exposed ice have occurred in seasons with relatively little snow in late spring (2022) and in seasons with above average spring snow depths (2024). A deep spring snow pack can undoubtedly delay the onset of ice ablation and reduce annual ice loss, but recent years indicate that it cannot prevent substantial ice melt under otherwise unfavorable conditions.

[Figure]

*Winter and summer mass balance vs. annual mass balance at WGMS reference glaciers Vernagtferner and Hintereisferner.*

[Figure]

*Cleaned SR50 data showing surface height evolution at the AWS. The lower panel shows smoothed surface height and air temperature. This figure was added to the supplement in the revised manuscript and is referred to in comments on the role of winter snow pack for the ablation season added to the revised discussion section.*

For the References list there are no DOI numbers at any of the references. I assume TC would like some DOI numbers.
Added DOIs, thanks for pointing this out.

Section 2.4 needs some input on the spatial resolution of Sentinel 2 albedo and as well how the buffered pixels are handled. If multiple pixels are covering the buffer what is done ?
We have added the spatial resolution (10x10m) and an explanation of the buffer. The value for each point corresponds to the pixel whose centroid is within the buffer. If there are multiple pixel centroids in the buffer, the average is taken.

Ideally the configuration files for Cosipy and the modified code to account for albedo directly should be included in some repo for reproducibility.
Yes, we will prepare a repo to accompany the revised manuscript.

Somewhere in the introduction or in the description of the study site there should be a mention of the winter mass balance, how much it is in context to the melt. Even ballpark numbers if measurements are not available.
There are no measurements of winter mass balance from the study site. As mentioned above, annual mass balance in the region is driven by ablation and not strongly correlated

with winter mass balance. To give some more context on the general setting, we added the following to the study site section:
"This is in line with trends towards increasingly negative annual mass balance at three World Glacier Monitoring Service (WGMS) reference glaciers in close proximity to Gepatschferner (Hintereisferner, Vernagtferner, Kesselwandferner). Annual mass balance at these sites is strongly correlated with summer mass balance and trends in ablation dominate the overall mass balance trend in the region (WGMS, 2025). Winter mass balance at the reference glaciers shows no clear trend over time."

In figure 4 there is a very nice comparison of the observed and remotely sensed albedo. For me it seems to be or might be different comparing statistics between seasons or months. Overestimation during the melt seasons and underestimation during winter. Would it be valuable, also in relation to future studies that would like to adopt the methodology, to provide some statistics for this comparison on a monthly basis ? This could be an additional column to table 1.

S2 derived albedo is on average slightly lower than AWS albedo from October through December but this is not the case in the remaining winter months. Below are various statistics comparing the different months. We feel that this is not very conclusive regarding seasonal over- or underestimation and would prefer to keep the table in the manuscript as is. If the reviewer feels differently or has suggestions on other metrics to compute here we would be happy to explore this further.

| month | rms | Abs. mean bias | Mean difference | Median difference |
|---|---|---|---|---|
| 10 | 0.119 | 0.085 | -0.027 | -0.027 |
| 11 | 0.109 | 0.082 | -0.01 | -0.02 |
| 12 | 0.117 | 0.087 | -0.044 | -0.027 |
| 1 | 0.088 | 0.073 | 0.017 | 0.021 |
| 2 | 0.099 | 0.079 | 0.041 | 0.043 |
| 3 | 0.093 | 0.08 | 0.056 | 0.072 |
| 4 | 0.099 | 0.082 | 0.065 | 0.064 |
| 5 | 0.081 | 0.062 | 0.054 | 0.039 |
| 6 | 0.083 | 0.066 | 0.041 | 0.055 |
| 7 | 0.085 | 0.069 | 0.056 | 0.049 |
| 8 | 0.072 | 0.06 | 0.022 | 0.028 |
| 9 | 0.068 | 0.054 | 0.012 | 0.013 |

MINOR COMMENTS
L004: feedback or relationship would be better than connection
Changed to relationship

L007: remove conditions …low albedo values…

Changed as suggested.

L007: Recorded by what ? AWS or S2 ?

Changed to : "recorded by the weather station"

L008 Is this only ice ablation? Or just general ablation for the location ? Winter snow + firn + ice?

It is only ice ablation.

L016: The new Glambie paper in Nature would be a good additional reference here

Yes - added.

L021: More context is needed. For example: "…bare ice becomes exposed with lower surface albedo than firn" This is then referred to in the next sentence.

Changed to: "As snow lines rise and multi-year firn is depleted, bare ice becomes exposed. The newly exposed ice surfaces have a lower albedo than snow and firn, which affects the surface energy balance by increasing the amount of absorbed solar radiation, thereby creating a positive melt-albedo feedback"

L026: glacier albedo might be better here. Or snow and ice albedo. Same in L027

Changed to "overall glacier albedo and the albedo of bare ice have increasingly come into focus" - we specifically discuss ice albedo in the following sentences, hence we'd like to keep this distinction.

L039: Drive is not a good word here. Force or input data

Changed to force.

L056: …during extreme years… extreme years of what ? Rewrite this sentence so it is clear what is extreme.

Rephrased to simplify the sentence: "The continued loss of firn and reduced seasonal snow cover increases the relative importance of bare ice and the potential albedo variability of bare ice surfaces for glacier-wide albedo."

L057: In the following of what? In this study… could be used

Changed to "in this study".

L070: For non-native Alps people writin Gepatschferner glacier would be helpful

We added "Gepatsch Glacier" in parenthesis in the previous paragraph where the name is first introduced to clarify this. Ferner is a German word for glacier so "Gepatschferner glacier" translates to Gepatsch glacier glacier and we'd rather avoid that.

L071: I am a bit confused by this number, 0.05 km2, as the red area highlighted in Fig.1 showing the ice covered mountain seems to be more than ~1km ? Are you referring to the absolute peak of the mountain ? Perhaps, if this number is not important you could skip it to reduce confusion ? Is it possible that this is the area all the stakes are aggregated in fig 1 panel c ? Perhaps a small box there would be useful.

We removed the number. It is indeed confusing and not essential to anything that follows.

L073: what is cal BP ? Please write out.

Calibrated years before present - added to the text.

L073: change ice depth to ice thickness.

OK!

L095: Are they operated only during summer ? Redrilled in each spring. In L096 it is indicated the stakes are visited over the full year or is this inly for the summer. In the ablation stakes section (2.2) it might be clear if these stakes only survey summer mass balance or if the survey winter mass balance as well ?

The stakes are redrilled when they are close to melting out. They remain in the ice throughout the year and are only used for ablation measurements. We have added the following explanation here: "Seven ablation stakes were drilled on the Weißseespitze summit ice cap between 2017 and 2019. Another stake was added in 2022. The stakes consist of 2 m long wooden poles connected to each other with tubing to achieve total stake lengths of 6 to 8 m. The stakes are periodically redrilled when they are close to melting out" and clarify that winter mass balance is not measured at the end of the section.

L111: I would suggest to use SW_in and SW_out which I believe is pretty standard ? Or provide some insight in why ref is used. Reference or reflected ? On most other locations in the paper SWin and SWout is used. Please systematically go though the paper and make these the same in text and figures.

Included "reflected" here to clarify this and revised the manuscript to ensure consistent usage of terms.

L149: In this sentence "Multi-spectral reflectance was converted to broadband reflectance using the conversion developed by Liang (2001), following prior work addressing broadband albedo of glacier surfaces (Naegeli et al., 2017)." A bit more insight into what equation is used would be beneficial and what bands are input.

We have specified the input bands in the text: "Multi-spectral reflectance was converted to broadband reflectance using the conversion developed by Liang et al 2001, where broadband reflectance is a function of the blue, red, NIR, SWIR1 and SWIR2 bands."

The equation is:
A = 0.356* blue +0.130* red+0.373*NIR+0.085*SWIR1+0.072*SWIR2−0.0018
We can add it to the manuscript if that would be beneficial but it is the same as in the two references, so we feel that interested readers could refer to those works for further background.

L174: are => were and in L175
Changed the phrasing of the sentence to avoid this.

L179: the word model or simulate is missing after the citation, "to model the surface mass balance…"
Thanks, added missing word.

L182: take => use

Changed as suggested.

L186: To Cosipy, is the snowfall in height units or w.eq. ? Is there a conversion with density done ?
We have restructured this section to improve overall clarity. Snowfall is passed to cosipy in height units. A default conversion to density is done with a constant density for freshly fallen snow of 68 kg/m3. This has been added to the revised manuscript. We will add constants and the config file to a code repository that will accompany the revised manuscript.

L189: Could you explain a bit what idealised runs actually are ? Observed met data with 0.05 incremental changes in albedo to the full period from and to ? Running from fall to fall or during summer ? What is the timestep of the model runs needs to be added.
We have restructured and expanded this section to more clearly explain what was done. The model runs at hourly timesteps for the duration of the time periods used in the experiments, i.e. during summer.

L191-192: What does this sentence mean ? What assumptions for the subsurface are assumed ? Is something in the model skipped ? I would assume that there are some sort of initial conditions for the model ?
We have added information subsurface parameters that are set during initialization (ice thickness and bottom temperature). No part of the model is "skipped" in the sense of modifying the model code. Essentially, the key simplification is that we neglect precipitation (both rain and snow) and snow processes in the calculations and only consider summer conditions. We have rephrased and restructured this section for clarity. The main part now reads:
"We performed idealised sensitivity experiments to isolate the impact of albedo on melt in otherwise unchanged conditions by systematically varying albedo. The model was set up to highlight the influence of surface albedo on overall ablation, considering how "darker" vs. "brighter" bare ice influences mass balance, and how this varies with the time of year. To this end, we considered simplified summertime scenarios with a bare ice surface, omitting considerations of energy balance processes in a seasonal or multi-annual snowpack and energy input from precipitation. Albedo of the ice surface was kept constant over time and varied in increments of 0.05 for each run, so that each model run corresponds to one albedo value between 0.05 and 0.95. For high albedo values that would be indicative of snow, the assumption is that only a minimal amount of snow covers the bare ice, as would be the case for example during a small summer snow fall at high elevation. Subsurface assumptions for model initialization were an ice thickness of 6 m and a bottom temperature of -4°C."

L198: "However, running the model with input as recorded by the AWS for a sub period with SR50 data of acceptable quality shows good agreement between the modeled and measured surface height (Fig. S8, supplementary material) during the phase of snow melt

and subsequent ice ablation at the station." This sentence would improve by being re-written for clarity, it is a bit . Surface height should be surface height changes.
L198: For me it is a bit too simple to say: "Detailed validation exercises are beyond the scope of this study." The authors mention a good comparison to observed and modelled surface height changes which in a sense is a good calibration metric. Since there is a figure in the supplements please add some simple stats in the text of this agreement, i.e. rmse, R2 and bias for example.

We have restructured and expanded this section. It now includes a more extensive description of the evaluation procedure and statistics comparing modeled and observed surface change rates. The figure and accompanying text in the supplement was adapted to more clearly show the comparison and explain what was done and the limitations of this evaluation approach (i.e., noise in the SR50 data, no measurements of snow density).

L206: Add the years (2018 - 2024) in the text so it is clear that these are mean values for the whole study period.
Changed as suggested.
L211. Are these anomalies ? See comment for figure 3
Rephrased to match the updated figure.

L246: Cumulative ice ablation. This is summer mass balance right ? No only ice, but ice and snow is it not ? Or skip the word "ice"
No, we do actually mean ice ablation. This is further explained in section 2.2 Ablation Stakes. In most cases the ice ablation values are equivalent to annual mass balance but we would like to keep the distinction because we do not explicitly account for snow at the stake locations.

L256: is => was
We think "is" is appropriate here because it is a "time invariant" statement but we are also not english language experts. In the past we have had great support from the language and copy editing team of copernicus journals so perhaps this is something they or the editor could decide. We are happy to correct it if needed and learn how to do this properly.

L261: mm w.e => mm w.e. (a dot is missing)
Fixed, thanks!

L353: "broadband conversion of Liang (2001) is suitable for ice albedo." Not for snow albedo ? Perhaps expand this a bit so it is clear.
Changed to "glacier albedo" to clarify. Naegeli et al (2017)'s findings, which we refer to here, apply to ice and snow on glaciers.

L373: observation => observations

Fixed!

Figure 1: In the map (b) of Gepatschferner there are a lot of green outlines. I would suggest highlighting what is defined as Gepatschferner with another color for context and then either skip the other outlines or have them in an alternative color. A lot of ice patches and small glaciers are in the area. Also, there are parts of glaciers that seem to have no outline, i.e, in the south west part of the image. It also seems like parts of the Gepatschferner are missing its outline ?

Thanks, agreed. We have removed all outlines except for that of Gepatschferner and changed the color and line width to improve visibility. The parts that look like they are missing an outline (southwest section) drain towards the south into the neighboring watershed. The glacier boundary follows the ice divide and this section is not technically considered part of Gepatschferner. We feel that it is valuable to show it anyway since it is obviously part of the same ice mass but adhere to the boundary as per the regional inventory to maintain clear definitions of which part belongs to which glacier.

Figure 3. I am a bit confused by the figure. I understand that the long term mean albedo is shown in each subplot but for each individual year are the anomalies shown (data minus the long term mean) or the change compared to the mean ? For example, for the big grey period in 2024 in September is the anomaly about 0.4 lower than the long term mean or is the observed albedo about 0.2 ? I would change this figure to true anomalies (data minus mean). That would also highlight better the deviations as positive or negative. The AWS and S2 validation/comparison is already done in Figure 4 and 5 so there is no need to repeat that here with the black dots.

We have adapted the figure to improve clarity as follows:
Removed black dots showing S2, visually highlighted daily anomalies (black line) and reduced focus on the time series mean (grey line), adapted the caption to more clearly explain what is shown in the figure. We are aiming to show both the daily albedo as measured (black line) and indicate periods with below or above normal albedo (grey and blue shading). Daily anomalies are the vertical extent of the shading. The y-axis shows absolute albedo, as is now stated more clearly in the caption.

Figure 7: In the b-panel the color needs to be different for SWin and SWout. It is very hard to see what is what. Perhaps change to SWnet ? Or have SWin and SWout with different sign as often is done?
Changed the panel to show SWnet, thanks for the suggestion.

Figure 8 is a very nice figure.
Thank you!

Figure 10: It would be a good idea to have the same names for radiation components as mentioned earlier, SW/LW in/out here but different earlier in the text.

Changed the text to ensure consistency.

Figure 11: Meam should be mean

Fixed, thanks!

---

## Referee Report (RR1)

Thank you for the detailed responses and the thoughtful revisions made to the manuscript, which clearly clarify and fully address the suggestions raised during the review. I have just a *few minor points of clarification* following your comments and replies. Congratulations on this work, which presents very strong results and insightful discussions!

Spatial variability and albedo
3) Furthermore, regarding the results presented in Figure 12, it seems important to mention that the satellite images for 2023 and 2024 were acquired in September, a period when the glacier's albedo is likely not at its lowest (see Figure 2). Could the comparison of glacier-wide albedo between years be somewhat biased by the late acquisition dates in these two years (e.g., line 343)? This point could be addressed.

The images were selected for minimum snow cover and all suitable (cloud free) images were considered. Both 2023 and 2024 had relatively long ablation seasons that extended into October. In 2024, the lowest albedo at the AWS was measured on September 8 and the S2 image in Fig 12 is from Sep. 7 (see Fig 3 and Table 2). The image and the AWS minimum coincide very closely. In 2023, the lowest albedo value at the AWS occurred on August 23. A small snowfall event then brightened the surface. This snow melted again in the following days across most of the glacier, although some snow remained at the AWS location. The S2 image from August 24 is partially affected by clouds and cloud shadows (left image in the figure below). In the S2 image from September 10 (shown in Fig. 12 and below on the right) the remaining seasonal snow cover has retreated further compared to the Aug. 24 image, although a minimal amount of snow from the summer snowfall in late August remains at the AWS location. The exposed bare ice surface appears visually brighter in some areas of the Sep. 10 image but we do not believe this indicates a bias due to the time of year.

=> *Thank you for these details. This is indeed very interesting and would merit at least a brief mention in the revised manuscript.*

Line 83: Ice temperature sensor: Is it used in this study?

No. The information from the thermistor strings informs the initial ice temperature assumptions made in the energy balance model in a general way but this sensor is not essential for the study. We mention it here for completeness along with the other components of the AWS. We can remove this if it is confusing or seems unnecessary

=> *As these data are not used, I believe it would be more appropriate to remove them to avoid any confusion.*

Line 139 and throughout the document: "Low" and "very low" refer to specific values (i.e., 0.2 and 0.4) as indicated here. These terms are used throughout the document, sometimes with quotation marks and sometimes without. Conversely, "low albedo" is sometimes mentioned without explicitly referring to these values, making the text harder to follow. Please ensure that quotation marks are consistently used when referring to these specific values, or alternatively, use a uniform notation (e.g., alb < 0.2).

We will go through the manuscript to ensure consistency.

=> *Thanks. I still have a doubt regarding lines 193 and 195 (in the revised version) — should it be 'low' in quotation marks, or just low without them?*

Line 234, 238: "Generally coincide or occur" – This statement could be quantified (e.g., using delta day) to add more weight to the comparison.

We added a note on this in the revised manuscript. In 2018, S2-derived "low albedo" was observed five days prior to "low albedo" conditions at the AWS. Otherwise the S2-derived low albedo periods are within the AWS low albedo periods. In line 238, the statement is followed by a description of cases when there are discrepancies with examples. The stakes are not expected to have low albedo at exactly the same dates due to varying snow melt patterns. The irregular nature of the S2 time series makes it challenging to meaningfully interpret shifts of a few days - these may be due to snow melting earlier or later at one stake compared to the next, or to differing availability of S2 imagery.

=> *The second part of your answer is particularly interesting and relevant: (i.e.: "The stakes are not expected to have low albedo at exactly the same dates due to varying snow melt patterns. The irregular nature of the S2 time series makes it challenging to meaningfully interpret shifts of a few days - these may be due to snow melting earlier or later at one stake compared to the next, or to differing availability of S2 imagery.") In my opinion, this deserves to be included in the manuscript, even briefly, as it provides valuable context for interpreting the data.*

Technical corrections:
*line 304 (revised version): yr$^{-y}$-1 => yr$^{-1}$*
*line 593: MacDONELL => MacDonell*

---

## Author Response (AR3)

Responses to "technical corrections" by the editor:

The editor pointed out that "ablation" should not be a negative value since the word already implies that it is an ice loss. We have gone through the manuscript to use the term "surface mass balance" where the respective value has a negative sign, or change the sign when using "ablation".

L 81, 82: we removed a sentence part as suggested by editor and fixed parentheses in citation
L113: kept wording as is. The editor suggested "perpendicular to" but we feel that "plane with the ice surface" better reflects what we are trying to express. The stakes are drilled so that the end of the stake is more or less plane with the surface, i.e. not sticking out much. We are happy to change it if the editor or copy editors feel that perpendicular is better.

L292: Clarified following editor suggestion: "The stake positions are not expected to become snow free on exactly the same dates due to varying snow melt patterns"

L300: changed section heading to "Correlations between albedo and ablation in the WSS summit region" following editor suggestion
L312: shortened phrase following editor suggestion

Table 4: added a footnote to indicate the value affected by a data gap (editor suggested italics - this is also fine for us if better in keeping with journal format)

L523: Slightly changed this sentence to remove repetition following editor suggestion.